# Explicit Regularisation in Gaussian Noise Injections

**Alexander Camuto**
University of Oxford
Alan Turing Institute
acamuto@turing.ac.uk

**Matthew Willetts**
University of Oxford
Alan Turing Institute
mwilletts@turing.ac.uk

**Umut Şimşekli**
University of Oxford
Institut Polytechnique de Paris
umut.simsekli@telecom-paris.fr

**Stephen Roberts**
University of Oxford
Alan Turing Institute
sjrob@robots.ox.ac.uk

**Chris Holmes**
University of Oxford
Alan Turing Institute
cholmes@stats.ox.ac.uk

## Abstract

We study the regularisation induced in neural networks by Gaussian noise injections (GNIs). Though such injections have been extensively studied when applied to data, there have been few studies on understanding the regularising effect they induce when applied to network activations. Here we derive the explicit regulariser of GNIs, obtained by marginalising out the injected noise, and show that it penalises functions with high-frequency components in the Fourier domain; particularly in layers closer to a neural network's output. We show analytically and empirically that such regularisation produces calibrated classifiers with large classification margins.

## 1 Introduction

Noise injections are a family of methods that involve adding or multiplying samples from a noise distribution, typically an isotropic Gaussian, to the weights or activations of a neural network during training. The benefits of such methods are well documented. Models trained with noise often generalise better to unseen data and are less prone to overfitting (Srivastava et al., 2014; Kingma et al., 2015; Poole et al., 2014).

Even though the regularisation conferred by Gaussian noise injections (GNIs) can be observed empirically, and the benefits of noising data are well understood theoretically (Bishop, 1995; Cohen et al., 2019; Webb, 1994), there have been few studies on understanding the benefits of methods that inject noise *throughout* a network. Here we study the *explicit* regularisation of such injections, which is a positive term added to the loss function obtained when we marginalise out the noise we have injected.

Concretely our contributions are:

- We derive an analytic form for an explicit regulariser that explains most of GNIs' regularising effect.

- We show that this regulariser penalises networks that learn functions with high-frequency content in the Fourier domain and most heavily regularises neural network layers that are closer to the output. See Figure 1 for an illustration.

- Finally, we show analytically and empirically that this regularisation induces larger classification margins and better calibration of models.

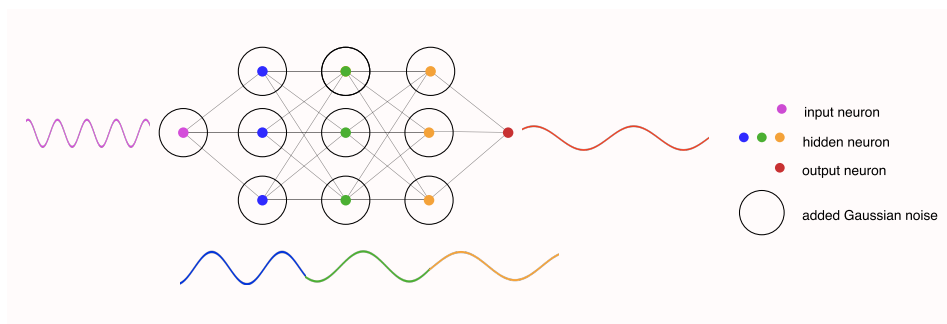

Figure 1: Here we illustrate the effect of GNIs injected throughout a network's activations. Each coloured dot represents a neuron's activations. We add GNIs, represented as circles, to each layer's activations bar the output layer. GNIs induce a network for which each layer learns a progressively lower frequency function, represented as a sinusoid matching in colour to its corresponding layer.

## 2 Background

### 2.1 Gaussian Noise Injections

Training a neural network involves optimising network parameters to maximise the marginal likelihood of a set of labels given features via gradient descent. With a training dataset $\mathcal{D}$ composed of $N$ data-label pairs of the form $(\mathbf{x}, \mathbf{y})$ $\mathbf{x} \in \mathbb{R}^d, \mathbf{y} \in \mathbb{R}^m$ and a feed-forward neural network with $M$ parameters divided into $L$ layers: $\boldsymbol{\theta} = \{\mathbf{W}_1, ..., \mathbf{W}_L\}$, $\boldsymbol{\theta} \in \mathbb{R}^M$, our objective is to minimise the expected negative log likelihood of labels $\mathbf{y}$ given data $\mathbf{x}$, $-\log p_{\boldsymbol{\theta}}(\mathbf{y}|\mathbf{x})$ , and find the optimal set of parameters $\boldsymbol{\theta}^*$ satisfying:

$$\boldsymbol{\theta}_* = \arg\min_{\boldsymbol{\theta}} \mathcal{L}(\mathcal{D}; \boldsymbol{\theta}), \qquad \mathcal{L}(\mathcal{D}; \boldsymbol{\theta}) := -\mathbb{E}_{\mathbf{x}, \mathbf{y} \sim \mathcal{D}} \left[\log p_{\boldsymbol{\theta}}(\mathbf{y}|\mathbf{x})\right] . \tag{1}$$

Under stochastic optimisation algorithms, such as Stochastic Gradient Descent (SGD), we estimate $\mathcal{L}$ by sampling a mini-batch of data-label pairs $\mathcal{B} \subset \mathcal{D}$.

$$\mathcal{L}(\mathcal{B}; \boldsymbol{\theta}) = -\mathbb{E}_{\mathbf{x}, \mathbf{y} \sim \mathcal{B}} \log p_{\boldsymbol{\theta}}(\mathbf{y}|\mathbf{x}) \approx \mathcal{L}(\mathcal{D}; \boldsymbol{\theta}). \tag{2}$$

Consider an $L$ layer network with no noise injections and a non-linearity $\phi$ at each layer. We obtain the activations $\mathbf{h} = \{\mathbf{h}_0, ..., \mathbf{h}_L\}$, where $\mathbf{h}_0 = \mathbf{x}$ is the input data *before* any noise is injected. For a network consisting of dense layers (a.k.a. a multi-layer perceptron: MLP) we have that:

$$\mathbf{h}_k(\mathbf{x}) = \phi(\mathbf{W}_k \mathbf{h}_{k-1}(\mathbf{x})). \tag{3}$$

What happens to these activations when we inject noise? First, let $\boldsymbol{\epsilon}$ be the set of noise injections at each layer: $\boldsymbol{\epsilon} = \{\boldsymbol{\epsilon}_0, ..., \boldsymbol{\epsilon}_{L-1}\}$. When performing a noise injection procedure, the value of the next layer's activations depends on the noised value of the previous layer. We denote the intermediate, soon-to-be-noised value of an activation as $\widehat{\mathbf{h}}_k$ and the subsequently noised value as $\widetilde{\mathbf{h}}_k$:

$$\widehat{\mathbf{h}}_k(\mathbf{x}) = \phi\left(\mathbf{W}_k \widetilde{\mathbf{h}}_{k-1}(\mathbf{x})\right), \qquad \widetilde{\mathbf{h}}_k(\mathbf{x}) = \widehat{\mathbf{h}}_k(\mathbf{x}) \circ \boldsymbol{\epsilon}_k , \tag{4}$$

where $\circ$ is some element-wise operation. We can, for example, add or multiply Gaussian noise to each hidden layer unit. In the additive case, we obtain:

$$\widetilde{\mathbf{h}}_k(\mathbf{x}) = \widehat{\mathbf{h}}_k(\mathbf{x}) + \boldsymbol{\epsilon}_k, \qquad \boldsymbol{\epsilon}_k \sim \mathcal{N}(0, \sigma_k^2 \mathbf{I}). \tag{5}$$

The multiplicative case can be rewritten as an activation-scaled addition:

$$\widetilde{\mathbf{h}}_k(\mathbf{x}) = \widehat{\mathbf{h}}_k(\mathbf{x}) + \boldsymbol{\epsilon}_k, \qquad \boldsymbol{\epsilon}_k \sim \mathcal{N}\left(0, \widehat{\mathbf{h}}_k^2(\mathbf{x})\sigma_k^2 \mathbf{I}\right). \tag{6}$$

Here we focus our analysis on noise *additions*, but through equation (6) we can translate our results to the multiplicative case.

### 2.2 Sobolev Spaces

To define a Sobolev Space we use the generalisation of the derivative for multivariate functions of the form $g : \mathbb{R}^d \to \mathbb{R}$. We use a multi-index notation $\alpha \in \mathbb{R}^d$ which defines mixed partial derivatives. We denote the $\alpha^{\text{th}}$ derivative of $g$ with respect to its input $\mathbf{x}$ as $D^\alpha g(\mathbf{x})$.

$$D^\alpha g = \frac{\partial^{|\alpha|} g}{\partial x_1^{\alpha_1} \ldots \partial x_d^{\alpha_d}}$$

where $|\alpha| = \sum_{i=1}^d |\alpha_i|$. Note that $\mathbf{x}^\alpha = [x_1^{\alpha_1}, \ldots, x_d^{\alpha_d}]$ and $\alpha! = \alpha_1! \cdot \cdots \cdot \alpha_d!$.

**Definition 2.1** ([Cucker and Smale (2002)](#)). *Sobolev spaces are denoted $W^{l,p}(\Omega), \Omega \subset \mathbb{R}^d$, where $l$, the order of the space, is a non-negative integer and $p \geq 1$. The Sobolev space of index $(l,p)$ is the space of locally integrable functions $f : \Omega \to \mathbb{R}$ such that for every multi-index $\alpha$ where $|\alpha| < l$ the derivative $D^\alpha f$ exists and $D^\alpha f \in L^p(\Omega)$. The norm in such a space is given by* $\|f\|_{W^{l,p}(\Omega)} = \left( \sum_{|\alpha| \leq l} \int_\Omega |D^\alpha f(\mathbf{x})|^p d\mathbf{x} \right)^{\frac{1}{p}}$.

For $p = 2$ these spaces are Hilbert spaces, with a dot product that defines the $L_2$ norm of a function's derivatives. Further these Sobolev spaces can be defined in a measure space with *finite* measure $\mu$. We call such spaces finite measure spaces of the form $W_\mu^{l,p}(\mathbb{R}^d)$ and these are the spaces of locally integrable functions such that for every $\alpha$ where $|\alpha| < l$, $D^\alpha f \in L_\mu^p(\mathbb{R}^d)$, the $L^p$ space equipped with the measure $\mu$. The norm in such a space is given by ([Hornik, 1991](#)):

$$\|f\|_{W_\mu^{l,p}(\mathbb{R}^d)} = \left( \sum_{|\alpha| \leq l} \int_{\mathbb{R}^d} |D^\alpha f(\mathbf{x})|^p d\mu(\mathbf{x}) \right)^{\frac{1}{p}}, f \in W_\mu^{l,p}(\mathbb{R}^d), |\mu(\mathbf{x})| < \infty \; \forall \mathbf{x} \in \mathbb{R}^d \quad (7)$$

Generally a Sobolev space over a compact subset $\Omega$ of $\mathbb{R}^d$ can be expressed as a weighted Sobolev space with a measure $\mu$ which has compact support on $\Omega$ ([Hornik, 1991](#)).

[Hornik (1991)](#) have shown that neural networks with continuous activations, which have continuous and bounded derivatives up to order $l$, such as the sigmoid function, are universal approximators in the *weighted* Sobolev spaces of order $l$, meaning that they form a dense subset of Sobolev spaces. Further, [Czarnecki et al. (2017)](#) have shown that networks that use piecewise linear activation functions (such as ReLU and its extensions) are *also* universal approximators in the Sobolev spaces of order 1 where the domain $\Omega$ is some compact subset of $\mathbb{R}^d$. As mentioned above, this is equivalent to being dense in a weighted Sobolev space on $\mathbb{R}^d$ where the measure $\mu$ has compact support. Hence, we can view a neural network, with sigmoid or piecewise linear activations to be a parameter that indexes a function in a weighted Sobolev space with index $(1,2)$, i.e. $f_{\boldsymbol{\theta}} \in W_\mu^{1,2}(\mathbb{R}^d)$.

# 3 The Explicit Effect of Gaussian Noise Injections

Here we consider the case where we noise all layers with *isotropic* noise, except the final predictive layer which we also consider to have no activation function. We can express the effect of the Gaussian noise injection on the cost function as an added term $\Delta\mathcal{L}$, which is dependent on $\boldsymbol{\mathcal{E}}_L$, the noise accumulated on the final layer $L$ from the noise additions $\boldsymbol{\epsilon}$ on the previous hidden layer activations.

$$\widetilde{\mathcal{L}}(\mathcal{B}; \boldsymbol{\theta}, \boldsymbol{\epsilon}) = \mathcal{L}(\mathcal{B}; \boldsymbol{\theta}) + \Delta\mathcal{L}(\mathcal{B}; \boldsymbol{\theta}, \boldsymbol{\mathcal{E}}_L) \quad (8)$$

To understand the regularisation induced by GNIs, we want to study the regularisation that these injections induce *consistently* from batch to batch. To do so, we want to remove the stochastic component of the GNI regularisation and extract a regulariser that is of consistent sign. Regularisers that change sign from batch-to-batch do not give a constant objective to optimise, making them unfit as regularisers ([Botev et al., 2017](#); [Sagun et al., 2018](#); [Wei et al., 2020](#)).

As such, we study the explicit regularisation these injections induce by way of the expected regulariser, $\mathbb{E}_{\boldsymbol{\epsilon} \sim p(\boldsymbol{\epsilon})} [\Delta\mathcal{L}(\cdot)]$ that marginalises out the injected noise $\boldsymbol{\epsilon}$. To lighten notation, we denote this as $\mathbb{E}_{\boldsymbol{\epsilon}} [\Delta\mathcal{L}(\cdot)]$. We extract $R$, a constituent term of the expected regulariser that dominates the remainder terms in norm, and is *consistently positive*.

Because of these properties, $R$ provides a lens through which to study the effect of GNIs. As we show, this term has a connection to the Sobolev norm and the Fourier transform of the function parameterised by the neural network. Using these connections we make inroads into better understanding the regularising effect of noise injections on neural networks.

To begin deriving this term, we first need to define the accumulated noise $\boldsymbol{\mathcal{E}}_L$. We do so by applying a Taylor expansion to each noised layer. As in Section [2.2](#) we use the generalisation of the derivative for multivariate functions using a multi-index $\alpha$. For example $D^\alpha h_{k,i}(\mathbf{h}_{k-1}(\mathbf{x}))$ denotes the $\alpha^{\text{th}}$ derivative of the $i^{\text{th}}$ activation of the $k^{\text{th}}$ layer $(h_{k,i})$ with respect to the preceding layer's activations $\mathbf{h}_{k-1}(\mathbf{x})$ and $D^\alpha \mathcal{L}(\mathbf{h}_k(\mathbf{x}), \mathbf{y})$ denotes the $\alpha^{\text{th}}$ derivative of the loss with respect to the non-noised activations $\mathbf{h}_k(\mathbf{x})$.

**Proposition 1.** *Consider an $L$ layer neural network experiencing isotropic GNIs at each layer $k \in [0, \ldots, L-1]$ of dimensionality $d_k$. We denote this added noise as $\boldsymbol{\epsilon} = \{\boldsymbol{\epsilon}_0, ..., \boldsymbol{\epsilon}_{L-1}\}$. We assume $\mathbf{h}_L$ is in $C^\infty$ the class of infinitely differentiable functions. We can define the accumulated noise at layer each layer $k$ using a multi-index $\alpha_k \in \mathbb{N}^{d_{k-1}}$:*

$$\mathcal{E}_{L,i} = \sum_{|\alpha_L|=1}^{\infty} \frac{1}{\alpha_L!} \left(D^{\alpha_L} h_{L,i}(\mathbf{h}_{L-1}(\mathbf{x}))\right) \boldsymbol{\mathcal{E}}_{L-1}^{\alpha_L}, \; i = 1, \ldots, d_L$$

$$\mathcal{E}_{k,i} = \epsilon_{k,i} + \sum_{|\alpha_k|=1}^{\infty} \frac{1}{\alpha_k!} \left(D^{\alpha_k} h_{k,i}(\mathbf{h}_{k-1}(\mathbf{x}))\right) \boldsymbol{\mathcal{E}}_{k-1}^{\alpha_k}, \; i = 1, \ldots, d_k, \; k = 1 \ldots L-1$$

$$\boldsymbol{\mathcal{E}}_0 = \boldsymbol{\epsilon}_0$$

*where $\mathbf{x}$ is drawn from the dataset $\mathcal{D}$, $\mathbf{h}_k$ are the activations before any noise is added, as defined in Equation (3).*

See Appendix A.1 for the proof. Given this form for the accumulated noise, we can now define the expected regulariser. For compactness of notation, we denote each layer's Jacobian as $\mathbf{J}_k \in \mathbb{R}^{d_L \times d_k}$ and the Hessian of the loss with respect to the final layer as $\mathbf{H}_L \in \mathbb{R}^{d_L \times d_L}$. Each entry of $\mathbf{J}_k$ is a partial derivative of $f_{k,i}^\theta$, the function from layer $k$ to the $i^{\text{th}}$ network output, $i = 1...d_L$.

$$\mathbf{J}_k(\mathbf{x}) = \begin{bmatrix} \frac{f_{k,1}^\theta}{\partial h_{k,1}} & \frac{f_{k,1}^\theta}{\partial h_{k,2}} & \cdots \\ \vdots & \ddots & \\ \frac{f_{k,d_L}^\theta}{\partial h_{k,1}} & & \frac{f_{k,d_L}^\theta}{\partial h_{k,d_k}} \end{bmatrix}, \; \mathbf{H}_L(\mathbf{x}, \mathbf{y}) = \begin{bmatrix} \frac{\partial^2 \mathcal{L}}{\partial h_{L,1}^2} & \frac{\partial^2 \mathcal{L}}{\partial h_{L,1} \partial h_{L,2}} & \cdots \\ \vdots & \ddots & \\ \frac{\partial^2 \mathcal{L}}{\partial h_{L,d_L} \partial h_{L,1}} & & \frac{\partial^2 \mathcal{L}}{\partial h_{L,d_L}^2} \end{bmatrix}$$

Using these notations we can now define the explicit regularisation induced by GNIs.

**Theorem 1.** *Consider an $L$ layer neural network experiencing isotropic GNIs at each layer $k \in [0, \ldots, L-1]$ of dimensionality $d_k$. We denote this added noise as $\boldsymbol{\epsilon} = \{\boldsymbol{\epsilon}_0, ..., \boldsymbol{\epsilon}_{L-1}\}$. We assume $\mathcal{L}$ is in $C^\infty$ the class of infinitely differentiable functions. We can marginalise out the injected noise $\boldsymbol{\epsilon}$ to obtain an added regulariser:*

$$\mathbb{E}_{\boldsymbol{\epsilon}} \left[\Delta \mathcal{L}(\mathcal{B}; \boldsymbol{\theta}, \boldsymbol{\mathcal{E}}_L)\right] = \mathbb{E}_{(\mathbf{x},\mathbf{y}) \sim \mathcal{B}} \left[\frac{1}{2} \sum_{k=0}^{L-1} \left[\sigma_k^2 \text{Tr}\left(\mathbf{J}_k^\intercal(\mathbf{x}) \mathbf{H}_L(\mathbf{x}, \mathbf{y}) \mathbf{J}_k(\mathbf{x})\right)\right]\right] + \mathbb{E}_{\boldsymbol{\epsilon}} \left[\mathcal{C}(\mathcal{B}, \boldsymbol{\epsilon})\right]$$

*where $\mathbf{h}_k$ are the activations before any noise is added, as in equation (3). $\mathbb{E}_{\boldsymbol{\epsilon}} \left[\mathcal{C}(\cdot)\right]$ is a remainder term in higher order derivatives.*

See Appendix A.2 for the proof and for the exact form of the remainder. We denote the first term in Theorem 1 as:

$$R(\mathcal{B}; \boldsymbol{\theta}) = \mathbb{E}_{(\mathbf{x},\mathbf{y}) \sim \mathcal{B}} \left[\frac{1}{2} \sum_{k=0}^{L-1} \left[\sigma_k^2 \text{Tr}\left(\mathbf{J}_k^\intercal(\mathbf{x}) \mathbf{H}_L(\mathbf{x}, \mathbf{y}) \mathbf{J}_k(\mathbf{x})\right)\right]\right] \quad (9)$$

To understand the main contributors behind the regularising effect of GNIs, we first want to establish the relative importance of the two terms that constitute the explicit effect. We know that $R$ is the added regulariser for the linearised version of a neural network, defined by its Jacobian. This linearisation well approximates neural network behaviour for *sufficiently wide* networks (Jacot et al., 2018; Chizat et al., 2019; Arora et al., 2019) in early stages of training (Chen et al., 2020), and we can expect $R$ to dominate the remainder term in norm, which consists of higher order derivatives. In Figure 2 we show that this is the case for a range of GNI variances, datasets, and activation functions for networks with 256 neurons per layer; where the remainder is estimated as:

$$\mathbb{E}_{\boldsymbol{\epsilon}} \left[\mathcal{C}(\mathcal{B}, \boldsymbol{\epsilon})\right] \approx \frac{1}{1000} \sum_{i=0}^{1000} \widetilde{\mathcal{L}}(\mathcal{B}; \boldsymbol{\theta}, \boldsymbol{\epsilon}) - R(\mathcal{B}; \boldsymbol{\theta}) - \mathcal{L}(\mathcal{B}; \boldsymbol{\theta}).$$

These results show that $R$ is a significant component of the regularising effect of GNIs. It dominates the remainder $\mathbb{E}_{\boldsymbol{\epsilon}} \left[\mathcal{C}(\cdot)\right]$ in norm and is always positive, as we will show, thus offering a consistent objective for SGD to minimise. Given that $R$ is a likely candidate for understanding the effect of GNIs; we further study this term separately in regression and classification settings.

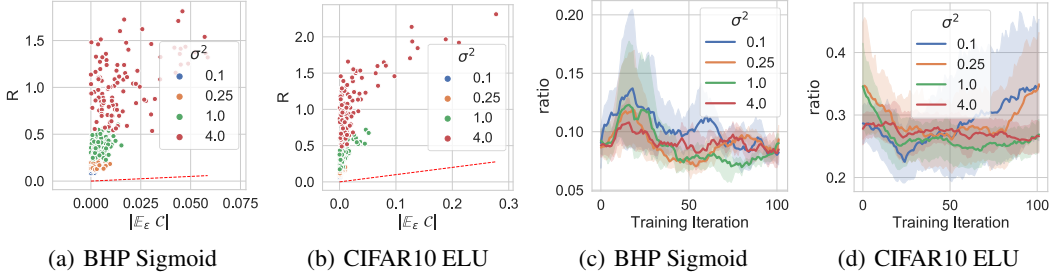

(a) BHP Sigmoid      (b) CIFAR10 ELU      (c) BHP Sigmoid      (d) CIFAR10 ELU

Figure 2: In (a,b) we plot $R(\cdot)$ vs $\mathbb{E}_{\epsilon}[\mathcal{C}(\cdot)]$ at initialisation for 6-layer-MLPs with GNIs at each 256-neuron layer with the same variance $\sigma^2 \in [0.1, 0.25, 1.0, 4.0]$ at each layer. Each point corresponds to one of 250 different network initialisation acting on a batch of size 32 for the classification dataset CIFAR10 and regression dataset Boston House Prices (BHP) datasets. The dotted red line corresponds to $y = x$ and demonstrates that for all batches and GNI variances $R$ is greater than $\mathbb{E}_{\epsilon}[\mathcal{C}(\cdot)]$. In (c,d) we plot ratio $= |\mathbb{E}_{\epsilon}[\mathcal{C}(\cdot)]|/R(\cdot)$ in the first 100 training iteration for 10 randomly initialised networks. Shading corresponds to the standard deviation of values over the 10 networks. $R(\cdot)$ remains dominant in early stages of training as the ratio is less than 1 for all steps.

**Regularisation in Regression** In the case of regression one of the most commonly used loss functions is the mean-squared error (MSE), which is defined for a data label pair $(\mathbf{x}, \mathbf{y})$ as:

$$\mathcal{L}(\mathbf{x}, \mathbf{y}) = \frac{1}{2}(\mathbf{y} - \mathbf{h}_L(\mathbf{x}))^2. \tag{10}$$

For this loss, the Hessians in Theorem 1 are simply the identity matrix. The explicit regularisation term, guaranteed to be positive is:

$$R(\mathcal{B}; \boldsymbol{\theta}) = \frac{1}{2}\mathbb{E}_{\mathbf{x} \sim \mathcal{B}} \left[ \sum_{k=0}^{L-1} \sigma_k^2 (\|\mathbf{J}_k(\mathbf{x})\|_F^2) \right]. \tag{11}$$

where $\sigma_k^2$ is the variance of the noise $\boldsymbol{\epsilon}_k$ injected at layer $k$ and $\|\cdot\|_F$ is the Frobenius norm. See Appendix A.4 for a proof.

**Regularisation in Classification** In the case of classification, we consider the case of a cross-entropy (CE) loss. Recall that we consider our network outputs $\mathbf{h}_L$ to be the pre-softmax of the logits of the final layer. We denote $\mathbf{p}(\mathbf{x}) = \mathrm{softmax}(\mathbf{h}_L(\mathbf{x}))$. For a pair $(\mathbf{x}, \mathbf{y})$ we have:

$$\mathcal{L}(\mathbf{x}, \mathbf{y}) = -\sum_{c=0}^{C} \mathbf{y}_c \log(\mathbf{p}(\mathbf{x}))_c), \tag{12}$$

where $c$ indexes over $C$ possible classes. The hessian $\mathbf{H}_L(\cdot)$ no longer depends on $\mathbf{y}$:

$$\mathbf{H}_L(\mathbf{x})_{i,j} = \begin{cases} \mathbf{p}(\mathbf{x})_i(1 - \mathbf{p}(\mathbf{x})_j) & i = j \\ -\mathbf{p}(\mathbf{x})_i \mathbf{p}(\mathbf{x})_j & i \neq j \end{cases} \tag{13}$$

This Hessian is positive-semi-definite and $R(\cdot)$, guaranteed to be positive, can be written as:

$$R(\mathcal{B}; \boldsymbol{\theta}) = \frac{1}{2}\mathbb{E}_{\mathbf{x} \sim \mathcal{B}} \left[ \sum_{k=0}^{L-1} \sigma_k^2 \sum_{i,j} (\mathrm{diag}(\mathbf{H}_L(\mathbf{x}))^{\mathsf{T}} \mathbf{J}_k^2(\mathbf{x}))_{i,j} \right], \tag{14}$$

where $\sigma_k^2$ is the variance of the noise $\boldsymbol{\epsilon}_k$ injected at layer $k$. See Appendix A.5 for the proof.

To test our derived regularisers, in Figure 3 we show that models trained with $R$ and GNIs have similar training profiles, whereby they have similar test-set loss and parameter Hessians throughout training, meaning that they have almost identical trajectories through the loss landscape. This implies that $R$ is a good descriptor of the effect of GNIs and that we can use this term to understand the mechanism underpinning the regularising effect of GNIs. As we now show, it penalises neural networks that parameterize functions with higher frequencies in the Fourier domain; offering a novel lens under which to study GNIs.

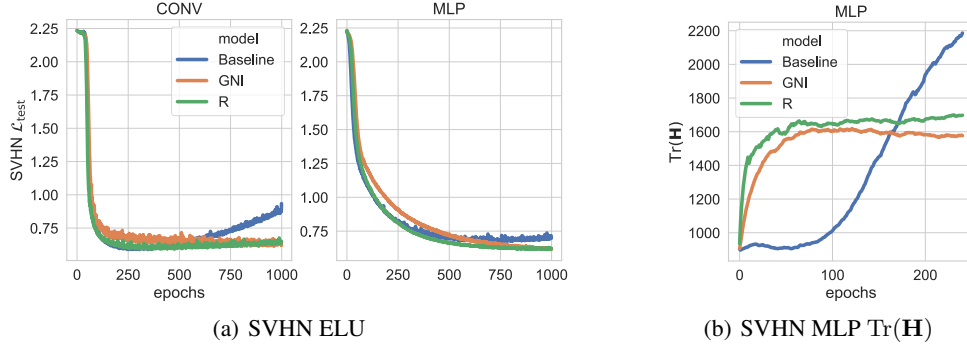

(a) SVHN ELU                                    (b) SVHN MLP $\mathrm{Tr}(\mathbf{H})$

Figure 3: Figure (a) shows the test set loss for convolutional models (CONV) and 4 layer MLPs trained on SVHN with $R(\cdot)$ and GNIs for $\sigma^2 = 0.1$, and no noise (Baseline). Figure (b) shows the trace of the network parameter Hessian for a 2-layer, 32-unit-per-layer MLP where $\mathbf{H}_{i,j} = \frac{\partial \mathcal{L}}{\partial w_i \partial w_j}$, which is a proxy for the parameters' location in the loss landscape. All networks use ELU activations. See Appendix F for more such results on other datasets and network architectures.

## 4   Fourier Domain Regularisation

To link our derived regularisers to the Fourier domain, we use the connection between neural networks and Sobolev Spaces mentioned above. Recall that by Hornik (1991), we can only assume a sigmoid or piecewise linear neural network parameterises a function in a weighted Sobolev space with measure $\mu$, if we assume that the measure $\mu$ has compact support on a subset $\Omega \in \mathbb{R}^d$. As such, we equip our space with the *probability* measure $\mu(\mathbf{x})$, which we assume has compact support on some subset $\Omega \subset \mathbb{R}^d$ where $\mu(\Omega) = 1$. We define it such that $d\mu(\mathbf{x}) = p(\mathbf{x})d\mathbf{x}$ where $d\mathbf{x}$ is the Lebesgue measure and $p(\mathbf{x})$ is the data density function. Given this measure, we can connect the derivative of functions that are in the Hilbert-Sobolev space $W_\mu^{1,2}(\mathbb{R}^d)$ to the Fourier domain.

**Theorem 2.** *Consider a function, $f : \mathbb{R}^d \to \mathbb{R}$, with a $d$-dimensional input and a single output with $f \in W_\mu^{1,2}(\mathbb{R}^d)$ where $\mu$ is a probability measure which we assume has compact support on some subset $\Omega \subset \mathbb{R}^d$ such that $\mu(\Omega) = 1$. Let us assume the derivative of $f$, $D^\alpha f$, is in $L^2(\mathbb{R}^d)$ for some multi-index $\alpha$ where $|\alpha| = 1$. Then we can write that:*

$$\sum_{|\alpha|=1} \|D^\alpha f\|_{L_\mu^2(\mathbb{R}^d)}^2 = \int_{\mathbb{R}^d} \sum_{j=1}^d \left| \mathcal{G}(\boldsymbol{\omega}, j)\overline{\mathcal{G}(\boldsymbol{\omega}, j) * \mathcal{P}(\boldsymbol{\omega})} \right| d\boldsymbol{\omega} \tag{15}$$

$$\mathcal{G}(\boldsymbol{\omega}, j) = \omega_j \mathcal{F}(\boldsymbol{\omega})$$

*where $\mathcal{F}$ is the Fourier transform of $f$, $\mathcal{P}$ is the Fourier transform or the 'characteristic function' of the probability measure $\mu$, $j$ indexes over $\boldsymbol{\omega} = [\omega_1, \ldots, \omega_d]$, $*$ is the convolution operator, and $\overline{(\cdot)}$ is the complex conjugate.*

See Appendix A.3 for the proof. Note that in the case where the dataset contains finitely many points, the integrals for the norms of the form $\|D^\alpha f_\theta\|_{L_\mu^2(\mathbb{R}^d)}^2$ are approximated by sampling a batch from the dataset which is distributed according to the presumed probability measure $\mu(\mathbf{x})$. Expectations over a batch thus approximate integration over $\mathbb{R}^d$ with the measure $\mu(\mathbf{x})$ and this approximation improves as the batch size grows. We can now use Theorem 2 to link $R$ to the Fourier domain.

**Regression**   Let us begin with the case of regression. Assuming differentiable and continuous activation functions, then the Jacobians within $R$ are equivalent to the derivatives in Definition 2.1. Theorem 2 only holds for functions that have 1-D outputs, but we can decompose the Jacobians $\mathbf{J}_k$ as the derivatives of multiple 1-D output functions. Recall, that the $i^{\text{th}}$ row of the matrix $\mathbf{J}_k$ is the set of partial derivatives of $f_{k,i}^\theta$, the function from layer $k$ to the $i^{\text{th}}$ network output, $i = 1 \ldots d_L$, with respect to the $k^{\text{th}}$ layer activations. Using this perspective, and the fact that each $f_{k,i}^\theta \in W_\mu^{1,2}(\mathbb{R}^{d_k})$ ($d_k$ is the dimensionality of the $k^{\text{th}}$ layer), if we assume that the probability measure of our space

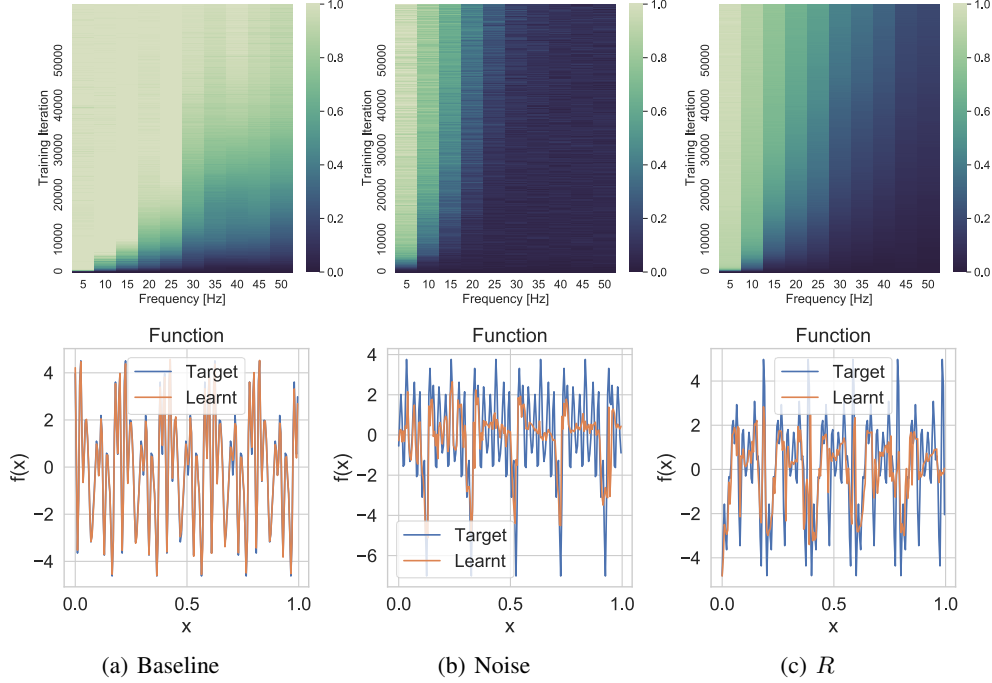

Figure 4: As in Rahaman et al. (2019), we train 6-layer deep 256-unit wide ReLU networks trained to regress the function $\lambda(z) = \sum_i \sin(2\pi r_i z + \phi(i))$ with $r_i \in (5, 10, \ldots, 45, 50)$. We train these networks with no noise (Baseline), with GNIs of variance 0.1 injected into each layer except the final layer (Noise), and with the $R(\cdot)$ for regression in (11). The first row shows the Fourier spectrum (x-axis) of the networks as training progresses (y-axis) averaged over 10 training runs. Colours show each frequency's amplitude clipped between 0 and 1. The second row shows samples of randomly generated target functions and the function learnt by the networks.

$\mu(\mathbf{x})$ has compact support, we use Theorem 2 to write:

$$
\begin{aligned}
R(\mathcal{B}; \boldsymbol{\theta}) &= \frac{1}{2}\mathbb{E}_{\mathbf{x}\sim\mathcal{B}}\left[\sum_{k=0}^{L-1}\sigma_k^2\sum_i\|\mathbf{J}_{k,i}(\mathbf{x})\|_F^2\right]\\
&= \frac{1}{2}\sum_{k=0}^{L-1}\sigma_k^2\sum_i\mathbb{E}_{\mathbf{x}\sim\mathcal{B}}\left[\|\mathbf{J}_{k,i}(\mathbf{x})\|_F^2\right]\\
&\approx \frac{1}{2}\sum_{k=0}^{L-1}\sigma_k^2\sum_i\sum_{|\alpha|=1}\|D^\alpha f_{k,i}^\theta\|_{L_\mu^2(\mathbb{R}^{d_k})}^2\\
&= \frac{1}{2}\sum_{k=0}^{L-1}\sigma_k^2\sum_i\int_{\mathbb{R}^{d_k}}\sum_{j=1}^{d_k}\left|\mathcal{G}_{k,i}^\theta(\boldsymbol{\omega},j)\overline{\mathcal{G}_{k,i}^\theta(\boldsymbol{\omega},j)*\mathcal{P}(\boldsymbol{\omega})}\right|d\boldsymbol{\omega} \quad (16)
\end{aligned}
$$

where $\mathbf{h}_0 = \mathbf{x}$, $i$ indexes over output neurons, and $\mathcal{G}_{k,i}^\theta(\boldsymbol{\omega},j) = \omega_j\mathcal{F}_{k,i}^\theta(\boldsymbol{\omega})$, where $\mathcal{F}_{k,i}^\theta$ is the Fourier transform of the function $f_{k,i}^\theta$. The approximation comes from the fact that in SGD, as mentioned above, integration over the dataset is approximated by sampling mini-batches $\mathcal{B}$.

If we take the data density function to be the empirical data density, meaning that it is supported on the $N$ points of the dataset $\mathcal{D}$ (i.e it is a set of $\delta$-functions centered on each point), then as the size $B$ of a batch $\mathcal{B}$ tends to $N$ we can write that:

$$
\lim_{B\to N}R(\mathcal{B};\boldsymbol{\theta}) = \frac{1}{2}\sum_{k=0}^{L-1}\sigma_k^2\sum_i\int_{\mathbb{R}^{d_k}}\sum_{j=1}^{d_k}\left|\mathcal{G}_{k,i}^\theta(\boldsymbol{\omega},j)\overline{\mathcal{G}_{k,i}^\theta(\boldsymbol{\omega},j)*\mathcal{P}(\boldsymbol{\omega})}\right|d\boldsymbol{\omega}. \quad (17)
$$

**Classification**   The classification setting requires a bit more work. Recall that our Jacobians are weighted by $\mathrm{diag}(\mathbf{H}_L(\mathbf{x}))^{\mathsf{T}}$, which has positive entries that are less than 1 by Equation (13). We can define a new set of measures such that $d\mu_i(\mathbf{x}) = \mathrm{diag}(\mathbf{H}_L(\mathbf{x}))_i^{\mathsf{T}} p(\mathbf{x}) d\mathbf{x}$, $i = 1 \ldots d_L$. Because this new measure is positive, finite and still has compact support, Theorem 2 still holds for the spaces indexed by $i$: $W_{\mu_i}^{1,2}(\mathbb{R}^d)$.

Using these new measures, and the fact that each $f_{k,i}^\theta \in W_{\mu_i}^{1,2}(\mathbb{R}^{d_k})$, we can use Theorem 2 to write that for classification models:

$$
\begin{aligned}
R(\mathcal{B}; \boldsymbol{\theta}) &= \frac{1}{2} \sum_{k=0}^{L-1} \sigma_k^2 \sum_i \mathbb{E}_{\mathbf{x} \sim \mathcal{B}} \left[ \mathrm{diag}(\mathbf{H}_L(\mathbf{x}))_i^{\mathsf{T}} \| \mathbf{J}_{k,i}(\mathbf{x}) \|_2^2 \right] \\
&\approx \frac{1}{2} \sum_{k=0}^{L-1} \sigma_k^2 \sum_i \sum_{|\alpha|=1} \| D^\alpha f_{k,i}^\theta \|_{L_{\mu_i}^2(\mathbb{R}^{d_k})}^2 \\
&= \frac{1}{2} \sum_{k=0}^{L-1} \sigma_k^2 \sum_i \int_{\mathbb{R}^{d_k}} \sum_{j=1}^{d_k} \left| \mathcal{G}_{k,i}^\theta(\boldsymbol{\omega}, j) \overline{\mathcal{G}_{k,i}^\theta(\boldsymbol{\omega}, j) * \mathcal{P}_i(\boldsymbol{\omega})} \right| d\boldsymbol{\omega}
\end{aligned}
\tag{18}
$$

Here $\mathcal{P}_i$ is the Fourier transform of the $i^{\text{th}}$ measure $\mu_i$ and as before $\mathcal{G}_{k,i}^\theta(\boldsymbol{\omega}, j) = \boldsymbol{\omega}_j \mathcal{F}_{k,i}^\theta(\boldsymbol{\omega})$, where $\mathcal{F}_{k,i}^\theta$ is the Fourier transform of the function $f_{k,i}^\theta$. Again as the batch size increases to the size of the dataset, this approximation becomes exact.

For both regression and classification, GNIs, by way of $R$, induce a prior which favours smooth functions with low-frequency components. This prior is enforced by the terms $\mathcal{G}_{k,i}^\theta(\boldsymbol{\omega}, j)$ which become large in magnitude when functions have high-frequency components, penalising neural networks that learn such functions. In Appendix B we also show that this regularisation in the Fourier domain corresponds to a form of *Tikhonov regularisation*.

In Figure 4, we demonstrate empirically that networks trained with GNIs learn functions that don't overfit; with lower-frequency components relative to their non-noised counterparts.

**A layer-wise regularisation**   Note that there is a recursive structure to the penalisation induced by $R$. Consider the layer-to-layer functions which map from a layer $k-1$ to $k$, $\mathbf{h}_k(\mathbf{h}_{k-1}(\mathbf{x}))$. With a slight abuse of notation, $\nabla_{\mathbf{h}_{k-1}} \mathbf{h}_k(\mathbf{x})$ is the Jacobian defined element-wise as:

$$
\left( \nabla_{\mathbf{h}_{k-1}} \mathbf{h}_k(\mathbf{x}) \right)_{i,j} = \frac{\partial h_{k,i}}{\partial h_{k-1,j}(\mathbf{x})},
$$

where as before $h_{k,i}$ is the $i^{\text{th}}$ activation of layer $k$.

$\| \nabla_{\mathbf{h}_{k-1}} \mathbf{h}_k(\mathbf{x}) \|_2^2$ is penalised $k$ times in $R$ as this derivative appears in $\mathbf{J}_0, \mathbf{J}_1 \ldots \mathbf{J}_{k-1}$ due to the chain rule. As such, when training with GNIs, we can expect the norm of $\| \nabla_{\mathbf{h}_{k-1}} \mathbf{h}_k(\mathbf{x}) \|_2^2$ to decrease as the layer index $k$ increases (i.e the closer we are to the network output). By Theorem 2, and Equations (16), and (18), larger $\| \nabla_{\mathbf{h}_{k-1}} \mathbf{h}_k(\mathbf{x}) \|_2^2$ correspond to functions with higher frequency components. Consequently, when training with GNIs the layer to layer function $\mathbf{h}_k(\mathbf{h}_{k-1}(\mathbf{x}))$ will have higher frequency components than the next layer's function $\mathbf{h}_{k+1}(\mathbf{h}_k(\mathbf{x}))$.

We measure this layer-wise regularisation in ReLU networks, using $\nabla_{\mathbf{h}_{k-1}} \mathbf{h}_k(\mathbf{x}) = \widetilde{\mathbf{W}}_k$. $\widetilde{\mathbf{W}}_k$ is obtained from the original weight matrix $\mathbf{W}_k$ by setting its $i^{\text{th}}$ column to zero whenever the neuron $i$ of the $k^{\text{th}}$ layer is inactive. Also note that the inputs of ReLU network hidden layers, which are the outputs of another ReLU-layer, will be positive. Negative weights are likely to 'deactivate' a ReLU-neuron, inducing smaller $\| \widetilde{\mathbf{W}}_k \|_2^2$, and thus parameterising a lower frequency function. We use the trace of a weight matrix as an indicator for the 'number' of negative components.

In Figure 5 we demonstrate that $\| \widetilde{\mathbf{W}}_k \|_2^2$ *and* $\mathrm{Tr}(\mathbf{W}_k)$ decrease as $k$ increases for ReLU-networks trained with GNIs, indicating that each successive layer in these networks learns a function with lower frequency components than the past layer. This striation and ordering is clearly absent in the baselines trained without GNIs.

**The Benefits of GNIs**   What does regularisation in the Fourier domain accomplish? The terms in $R$ are the traces of the Gauss-Newton decompositions of the second order derivatives of the loss. By

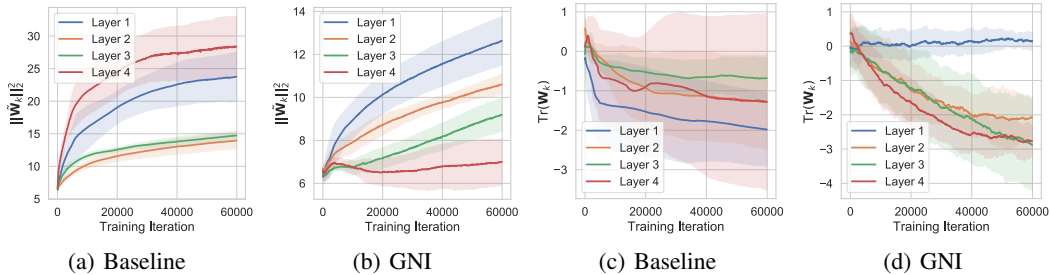

(a) Baseline      (b) GNI      (c) Baseline      (d) GNI

Figure 5: We use 6-layer deep 256-unit wide ReLU networks on the same dataset as in Figure 4 trained with (GNI) and without GNI (Baseline). In (a,b), for layers with square weight matrices, we plot $\|\widetilde{\mathbf{W}}_k\|_2^2$. In (c,d) we plot the trace of these layers' weight matrices $\mathrm{Tr}(\mathbf{W}_k)$. For GNI models, as the layer index $k$ increases, $\mathrm{Tr}(\mathbf{W}_k)$ and $\|\widetilde{\mathbf{W}}_k\|_2^2$ decrease, indicating that each successive layer in these networks learns a function with lower frequency components than the past layer.

penalising this we are more likely to land in wider (smoother) minima (see Figure 3), which has been shown, contentiously (Dinh et al., 2017), to induce networks with better generalisation properties (Keskar et al., 2019; Jastrzębski et al., 2017). GNIs however, confer other benefits too.

*Sensitivity.* A model's weakness to input perturbations is termed the *sensitivity*. Rahaman et al. (2019) have shown empirically that classifiers biased towards lower frequencies in the Fourier domain are less sensitive, and there is ample evidence demonstrating that models trained with noised data are less sensitive (Liu et al., 2019; Li et al., 2018). The Fourier domain - sensitivity connection can be established by studying the *classification margins* of a model (see Appendix D).

*Calibration.* Given a network's prediction $\hat{y}(\mathbf{x})$ with confidence $\hat{p}(\mathbf{x})$ for a point $\mathbf{x}$, perfect calibration consists of being as likely to be correct as you are confident: $p(\hat{y} = y|\hat{p} = r) = r, \ \forall r \in [0, 1]$ (Dawid, 1982; DeGroot and Fienberg, 1983). In Appendix E we show that models that are biased toward lower frequency spectra have lower 'capacity measures', which measure model complexity and lower values of which have been shown empirically to induce better calibrated models (Guo et al., 2017). In Figure F.7 we show that this holds true for models trained with GNIs.

## 5 Related Work

Many variants of GNIs have been proposed to regularise neural networks. Poole et al. (2014) extend this process and apply noise to all computational steps in a neural network layer. Not only is noise applied to the layer input it is applied to the layer output and to the pre-activation function logits. The authors allude to explicit regularisation but only derive a result for a single layer auto-encoder with a single noise injection. Similarly, Bishop (1995) derive an analytic form for the explicit regulariser induced by noise injections on *data* and show that such injections are equivalent to Tikhonov regularisation in an unspecified function space.

Recently Wei et al. (2020) conducted similar analysis to ours, dividing the effects of Bernoulli dropout into *explicit* and *implicit* effects. Their work is built on that of Mele and Altarelli (1993), Helmbold and Long (2015), and Wager et al. (2013) who perform this analysis for linear neural networks. Arora et al. (2020) derive an explicit regulariser for Bernoulli dropout on the final layer of a neural network. Further, recent work by Dieng et al. (2018) shows that noise additions on recurrent network hidden states outperform Bernoulli dropout in terms of performance and bias.

## 6 Conclusion

In this work, we derived analytic forms for the explicit regularisation induced by Gaussian noise injections, demonstrating that the explicit regulariser penalises networks with high-frequency content in Fourier space. Further we show that this regularisation is not distributed evenly within a network, as it disproportionately penalises high-frequency content in layers closer to the network output. Finally we demonstrate that this regularisation in the Fourier domain has a number of beneficial effects. It induces training dynamics that preferentially land in wider minima, it reduces model sensitivity to noise, and induces better calibration.

## Acknowledgments

This research was directly funded by the Alan Turing Institute under Engineering and Physical Sciences Research Council (EPSRC) grant EP/N510129/1. AC was supported by an EPSRC Studentship. MW was supported by EPSRC grant EP/G03706X/1. UŞ was supported by the French National Research Agency (ANR) as a part of the FBIMATRIX (ANR-16-CE23-0014) project. SR gratefully acknowledges support from the UK Royal Academy of Engineering and the Oxford-Man Institute. CH was supported by the Medical Research Council, the Engineering and Physical Sciences Research Council, Health Data Research UK, and the Li Ka Shing Foundation

## Impact Statement

This paper uncovers a new mechanism by which a widely used regularisation method operates and paves the way for designing new regularisation methods which take advantage of our findings. Regularisation methods produce models that are not only less likely to overfit, but also have better calibrated predictions that are more robust to distribution shifts. As such improving our understanding of such methods is critical as machine learning models become increasingly ubiquitous and embedded in decision making.

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
