[Supplementary Material]

# A Technical Proofs

## A.1 Proof of Proposition 1

*Proof of Proposition 1.* Recall that $\mathbf{h}$ denotes the vanilla activations of the network, those we obtain with no noise injection. Let us *not* inject noise in the final, predictive, layer of our network such that the noise on this layer is accumulated from the noising of previous layers.

We denote $\boldsymbol{\mathcal{E}}_k$ the noise accumulated at layer $k$ from GNIs in previous layers, and potential GNIs at layer $k$ itself. We denote $\mathcal{E}_{L,i}$ the $i^{\text{th}}$ element of the noise at layer $L$, the layer to which we do not add noise. This can be defined as a Taylor expansion around the accumulated noise at the previous layer $L-1$:

$$\mathcal{E}_{L,i} = \sum_{|\alpha_L|=1}^{\infty} \frac{1}{\alpha_L!} \left(D^{\alpha_L} h_{L,i}(\mathbf{h}_{L-1}(\mathbf{x}))\right) \boldsymbol{\mathcal{E}}_{L-1}^{\alpha_L} \tag{1}$$

where we use $\alpha_L$ as a multi-index over derivatives.

Generally if we noise all layers up to the penultimate layer of index $L-1$ we can define the accumulated noise at layer $k$, $\boldsymbol{\mathcal{E}}_k$ recursively because Gaussian have finite moments:

$$\mathcal{E}_{k,i} = \epsilon_{k,i} + \sum_{|\alpha_k|=1}^{\infty} \frac{1}{\alpha_k!} \left(D^{\alpha_k} h_{k,i}(\mathbf{h}_{k-1}(\mathbf{x}))\right) \boldsymbol{\mathcal{E}}_{k-1}^{\alpha_k}, \ i=1,\ldots,d_k, \ k=0\ldots L-1 \tag{2}$$

where $\boldsymbol{\mathcal{E}}_0 = \boldsymbol{\epsilon}_0$ is the base case.

$\square$

## A.2 Proof of Theorem 1

*Proof of Theorem 1.* Let us first consider the Taylor series expansion of the loss function with the accumulated noise defined in Proposition 1. Denoting $\boldsymbol{\epsilon} = [\boldsymbol{\epsilon}_{L-1}, \ldots, \boldsymbol{\epsilon}_0]$ we have:

$$\mathbb{E}_{\boldsymbol{\epsilon}}\left[\mathcal{L}(\mathbf{h}_L(\mathbf{x}) + \boldsymbol{\mathcal{E}}_L, \mathbf{y})\right] = \mathcal{L}(\mathbf{x}, \mathbf{y}) + \mathbb{E}_{\boldsymbol{\epsilon}}\left[\sum_{|\alpha|=1}^{\infty} \frac{1}{\alpha!} \left(D^{\alpha}\mathcal{L}(\mathbf{h}_L(\mathbf{x}), \mathbf{y})\right) \boldsymbol{\mathcal{E}}_L^{\alpha}\right] \tag{3}$$

Note that the dot product with the $i^{\text{th}}$ element of the final layer noise $\mathcal{E}_{L,i}$ can be written as

$$\mathbb{E}_{\boldsymbol{\epsilon}}\left[\sum_{|\alpha|=1}^{\infty} \frac{1}{\alpha!} \left(D^{\alpha}\mathcal{L}(\mathbf{h}_L(\mathbf{x}), \mathbf{y})\right) \mathcal{E}_{L,i}\right]$$

$$= \mathbb{E}_{\boldsymbol{\epsilon}}\left[\sum_{|\alpha|=1}^{\infty} \frac{1}{\alpha!} \left(D^{\alpha}\mathcal{L}(\mathbf{h}_L(\mathbf{x}), \mathbf{y})\right) \left(\sum_{|\alpha_L|=1}^{\infty} \frac{1}{\alpha_L!} \left(D^{\alpha_L} h_{L,i}(\mathbf{h}_{L-1}(\mathbf{x}))\right) \boldsymbol{\mathcal{E}}_{L-1}^{\alpha_L}\right)\right] \tag{4}$$

$$= \mathbb{E}_{\boldsymbol{\epsilon}}\left[\sum_{|\alpha|=1}^{\infty} \frac{1}{\alpha!} \left(D^{\alpha}\mathcal{L}(\mathbf{h}_L(\mathbf{x}), \mathbf{y})\right) \left(\sum_{|\alpha_L|=1}^{\infty} \frac{1}{\alpha_L!} \left(D^{\alpha_L} h_{L,i}(\mathbf{h}_{L-1}(\mathbf{x}))\right) (\boldsymbol{\epsilon}_{L-1} + \ldots)^{\alpha_L}\right)\right] \tag{5}$$

where the dots here denote the accumulated noise term on layer $L-1$ *before* we add the Gaussian noise $\boldsymbol{\epsilon}_{L-1}$. When looking at all elements of $\boldsymbol{\mathcal{E}}_L^{\alpha}$, not just the $i^{\text{th}}$ element, note that this is essentially the Taylor series expansion of $\mathcal{L}$ around the series expansion of $\mathbf{h}_L$ around $\boldsymbol{\epsilon}_{L-1}$. We know that the product of the Taylor series of a composed function $f \circ g$ with the Taylor series of $g$ is simply the Taylor series of $f$ around $\mathbf{x}$ (Constantine and Savits, 1996). This can be deduced from the slightly opaque Faà di Bruno's formula, which states that for multivariate derivatives of a composition of functions $f : \mathbb{R}^m \to \mathbb{R}$ and $g : \mathbb{R}^d \to \mathbb{R}^m$ and a multi-index $\alpha$ (Constantine and Savits, 1996)

$$D^{\alpha}(f \circ g)(\mathbf{x}) = \sum_{1 \leq |\lambda| \leq |\alpha|} D^{\lambda} f(g(\mathbf{x})) \sum_{s=1}^{|\alpha|} \sum_{p_s(\lambda, \alpha)} (\alpha!) \prod_{j=1}^{s} \frac{(D^{l_j} g(\mathbf{x}))^{k_j}}{(k_j!)[l_j!]^{|k_j|}},$$

where $p_s(\lambda, \alpha) = \{(k_1, \ldots, k_s); (l_1, \ldots, l_s) : |k_i| > 0, \ 0 \prec l_1 \cdots \prec l_s, \ \sum_{i=1}^{s} k_i = \lambda \sum_{i=1}^{s} |k_i| l_i = \alpha\}$, where $\prec$ denotes a partial order.

Applying this recursively to each layer $k$, we obtain that,

$$\mathbb{E}_{\epsilon}\left[\sum_{|\alpha|=1}^{\infty}\frac{1}{\alpha!}\left(D^{\alpha}\mathcal{L}(\mathbf{h}_L(\mathbf{x}),\mathbf{y})\right)\mathcal{E}_L^{\alpha}\right]$$

$$=\mathbb{E}_{\epsilon}\left[\sum_{k=0}^{L-1}\left[\sum_{|\alpha|=1}^{\infty}\frac{1}{\alpha_k!}\left(D^{\alpha_k}\mathcal{L}(\mathbf{h}_k(\mathbf{x}),\mathbf{y})\right)\epsilon_k^{\alpha_k}\right]+\mathcal{C}((\mathbf{x},\mathbf{y});\epsilon)\right] \quad (6)$$

Here $\mathcal{C}(\epsilon,\mathbf{x},\mathbf{y})$ represents cross-interactions between the noise at each layer $k$ $\epsilon_k$ and the noise injections at preceding layers with index less than $k$. We can further simplify the added term to the loss,

$$\mathbb{E}_{\epsilon}\left[\sum_{k=0}^{L-1}\left[\sum_{|\alpha|=1}^{\infty}\frac{1}{\alpha_k!}\left(D^{\alpha_k}\mathcal{L}(\mathbf{h}_k(\mathbf{x}),\mathbf{y})\right)\epsilon_k^{\alpha_k}\right]+\mathcal{C}((\mathbf{x},\mathbf{y});\epsilon)\right]$$

$$=\sum_{k=0}^{L-1}\left[\sum_{|\alpha_k|=1}^{\infty}\frac{1}{2\alpha_k!}\left(D^{2\alpha_k}\mathcal{L}(\mathbf{h}_k(\mathbf{x}),\mathbf{y})\right)\mathbb{E}_{\epsilon}\left[\epsilon_k^{2\alpha_k}\right]\right]+\mathbb{E}_{\epsilon}\left[\mathcal{C}((\mathbf{x},\mathbf{y});\epsilon)\right] \quad (7)$$

The second equality comes from the fact that odd-numbered moments of $\epsilon_k$, will be 0 and that $\mathcal{L}(\mathbf{x},\mathbf{y})=\mathcal{L}(\mathbf{h}_L(\mathbf{x}),\mathbf{y})$. The final equality comes from the moments of a mean 0 Gaussian, where $j$ takes the values of the multi-index. Note that $\left[\epsilon_k^{2\alpha_k}\right]$ are the even numbered moments of a zero mean Gaussian,

$$\mathbb{E}\left[\epsilon_k^{2\alpha_k}\right]=[\sigma_k^{2\alpha_{k,1}}(2\alpha_{k,1}-1)!,\ldots,\sigma_k^{2\alpha_{k,d_k}}(2\alpha_{k,d_k}-1)!]^{\mathsf{T}}$$

Though these equalities can already offer insight into the regularising mechanisms of GNIs, they are not easy to work with and will often be computationally intractable. We focus on the first set of terms here where each $|\alpha_k|=1$, which we denote $R(\mathbf{x},\boldsymbol{\theta})$

$$R(\mathbf{x},\theta)=\sum_{k=0}^{L-1}\left[\sum_{|\alpha_k|=1}\frac{1}{2\alpha_k!}\left(D^{2\alpha_k}\mathcal{L}(\mathbf{h}_k(\mathbf{x}),\mathbf{y})\right)\mathbb{E}_{\epsilon}\left[\epsilon_k^{2\alpha_k}\right]\right]$$

$$\approx\sum_{k=0}^{L-1}\left[\sum_{|\alpha_k|=1}\frac{\sigma_k^2}{2}\left(D^{2\alpha_k}\mathcal{L}(\mathbf{h}_L(\mathbf{x}),\mathbf{y})\right)\mathbf{J}_k^{2\alpha_k}(\mathbf{x})\right] \quad (8)$$

The last approximation corresponds to the Gauss-Newton approximation of second-order derivatives of composed functions where we've discarded the second set of terms of the form $D\mathcal{L}(\mathbf{h}_L)(\mathbf{x})\left(D^2\mathbf{h}_L(\mathbf{h}_k(\mathbf{x}))\right)$. We will include these terms in our remainder term $\mathcal{C}$. For compactness of notation, we denote each layer's Jacobian as $\mathbf{J}_k\in\mathbb{R}^{d_L\times d_k}$. Each entry of $\mathbf{J}_k$ is a partial derivative of $f_{k,i}^{\theta}$, the function from layer $k$ to the $i^{\text{th}}$ network output, $i=1...d_L$.

$$\mathbf{J}_k(\mathbf{x})=\begin{bmatrix}\frac{f_{k,1}^{\theta}}{\partial h_{k,1}}&\frac{f_{k,1}^{\theta}}{\partial h_{k,2}}&\cdots\\\vdots&\ddots&\\\frac{f_{k,d_L}^{\theta}}{\partial h_{k,1}}&&\frac{f_{k,d_L}^{\theta}}{\partial h_{k,d_k}}\end{bmatrix},$$

Again, for simplicity of notation $\mathbf{J}_k^{\alpha_k}$ selects the column indexed by $|\alpha_k|=1$. Also note that the sum over $|\alpha_k|=1$ effectively indexes over the diagonal of the Hessian of the Loss with respect to the $L^{\text{th}}$ layer activations. We denote this Hessian as $\mathbf{H}_L(\mathbf{x},\mathbf{y})\in\mathbb{R}^{d_L\times d_L}$.

$$\mathbf{H}_L(\mathbf{x},\mathbf{y})=\begin{bmatrix}\frac{\partial^2\mathcal{L}}{\partial h_{L,1}^2}&\frac{\partial^2\mathcal{L}}{\partial h_{L,1}\partial h_{L,2}}&\cdots\\\vdots&\ddots&\\\frac{\partial^2\mathcal{L}}{\partial h_{L,d_L}\partial h_{L,1}}&&\frac{\partial^2\mathcal{L}}{\partial h_{L,d_L}^2}\end{bmatrix}$$

This gives us that

$$R(\mathbf{x}; \boldsymbol{\theta}) = \frac{1}{2} \sum_{k=0}^{L-1} \left[ \sigma_k^2 \mathrm{Tr} \left( \mathbf{J}_k^{\mathsf{T}}(\mathbf{x}) \mathbf{H}_L(\mathbf{x}, \mathbf{y}) \mathbf{J}_k(\mathbf{x}) \right) \right]$$

For notational simplicity we include the terms that $R$ does not capture into the remainder $\mathbb{E}_{\boldsymbol{\epsilon}} \left[ \mathcal{C}((\mathbf{x}, \mathbf{y}); \boldsymbol{\epsilon}) \right]$. We take expectations over the batch and have:

$$\mathbb{E}_{(\mathbf{x}, \mathbf{y}) \sim \mathcal{B}} \left[ \mathbb{E}_{\boldsymbol{\epsilon}} \left[ \mathcal{L}(\mathbf{h}_L(\mathbf{x}) + \boldsymbol{\mathcal{E}}_L, \mathbf{y}) \right] \right] = \mathcal{L}(\mathcal{B}; \boldsymbol{\theta}) + R(\mathcal{B}; \boldsymbol{\theta}) + \mathbb{E}_{\boldsymbol{\epsilon}} \left[ \mathcal{C}(\mathcal{B}; \boldsymbol{\epsilon}) \right] \tag{9}$$

$$R(\mathcal{B}; \boldsymbol{\theta}) = \mathbb{E}_{(\mathbf{x}, \mathbf{y}) \sim \mathcal{B}} \left[ \frac{1}{2} \sum_{k=0}^{L-1} \left[ \sigma_k^2 \mathrm{Tr} \left( \mathbf{J}_k^{\mathsf{T}}(\mathbf{x}) \mathbf{H}_L(\mathbf{x}, \mathbf{y}) \mathbf{J}_k(\mathbf{x}) \right) \right] \right] \tag{10}$$

$$\mathbb{E}_{\boldsymbol{\epsilon}} \left[ \mathcal{C}(\mathcal{B}; \boldsymbol{\epsilon}) \right] = \mathbb{E}_{(\mathbf{x}, \mathbf{y}) \sim \mathcal{B}} \left[ \mathbb{E}_{\boldsymbol{\epsilon}} \left[ \sum_{|\alpha|=1}^{\infty} \frac{1}{\alpha!} \left( D^\alpha \mathcal{L}(\mathbf{h}_L(\mathbf{x}), \mathbf{y}) \right) \boldsymbol{\mathcal{E}}_L^\alpha \right] \right] - R(\mathcal{B}; \boldsymbol{\theta}) \tag{11}$$

This concludes the proof.

$\square$

### A.3 Proof of Theorem 2

*Proof of Theorem 2.* Because $f \in W_\mu^{1,2}(\mathbb{R}^d)$ we know that by definition, for $|\alpha| = 1$:

$$\| D^\alpha f \|_{L_\mu^2(\mathbb{R}^d)}^2 = \int_{\mathbb{R}^d} |D^\alpha f(\mathbf{x}) \cdot D^\alpha f(\mathbf{x}) \cdot \mu(\mathbf{x})| d\mathbf{x} < \infty$$

where $d\mathbf{x}$ is the Lebesgue measure. By Minkowski's inequality we know that:

$$\int_{\mathbb{R}^d} |D^\alpha f(\mathbf{x}) \cdot D^\alpha f(\mathbf{x}) \cdot \mu(\mathbf{x}) \cdot \mu(\mathbf{x})| d\mathbf{x} < \int_{\mathbb{R}^d} |\mu(\mathbf{x})| d\mathbf{x} \int_{\mathbb{R}^d} |D^\alpha f(\mathbf{x}) \cdot D^\alpha f(\mathbf{x}) \cdot \mu(\mathbf{x})| d\mathbf{x}$$

By definition $\mu$, a probability measure, is $L^1$ integrable. As such:

$$\int_{\mathbb{R}^d} |D^\alpha f(\mathbf{x}) \cdot D^\alpha f(\mathbf{x}) \cdot \mu(\mathbf{x}) \cdot \mu(\mathbf{x})| d\mathbf{x} < \infty$$

Let $m(\mathbf{x}) = D^\alpha f(\mathbf{x}) \cdot \mu(\mathbf{x})$, by the equation above, $m(\mathbf{x}) \in L^2(\mathbb{R}^d)$. As both $D^\alpha f(\mathbf{x})$ (by assumption) and $m(\mathbf{x})$ are $L^2$ integrable in $\mathbb{R}^d$, we can apply Fubini's Theorem and Plancherel's Theorem straighforwardly such that:

$$\sum_{|\alpha|=1} \| D^\alpha f \|_{L_\mu^2(\mathbb{R}^d)}^2 = \int_{\mathbb{R}^d} \sum_{j=1}^{d} \left| i \boldsymbol{\omega}_j \mathcal{F}(\boldsymbol{\omega}) \cdot \overline{\mathcal{M}(\boldsymbol{\omega}, j)} \right| d\boldsymbol{\omega}$$

where $\mathcal{F}$ is the Fourier transform of $f$, $i^2 = -1$, and $\boldsymbol{\omega}_j \mathcal{F}(\boldsymbol{\omega})$ is simply the Fourier transform of the derivative indexed by $\alpha$. $\mathcal{M}(\boldsymbol{\omega}, j)$ is given by

$$\mathcal{M}(\boldsymbol{\omega}, j) = i \left( \boldsymbol{\omega}_j \mathcal{F}(\boldsymbol{\omega}) \right) * \mathcal{P}(\boldsymbol{\omega}) \tag{12}$$

where $\mathcal{P}$ is the Fourier transform of the probability measure $\mu$, $\boldsymbol{\omega}_j \mathcal{F}(\boldsymbol{\omega})$ is as before, and $*$ denotes the convolution operator. Substituting $\mathcal{G}(\boldsymbol{\omega}, j) = \boldsymbol{\omega}_j \mathcal{F}(\boldsymbol{\omega})$ we obtain:

$$\sum_{|\alpha|=1} \| D^\alpha f \|_{L_\mu^2(\mathbb{R}^d)}^2 = \int_{\mathbb{R}^d} \sum_{j=1}^{d} \left| (i\bar{i}) \mathcal{G}(\boldsymbol{\omega}, j) \overline{\mathcal{G}(\boldsymbol{\omega}, j) * \mathcal{P}(\boldsymbol{\omega})} \right| d\boldsymbol{\omega}$$

$$= \int_{\mathbb{R}^d} \sum_{j=1}^{d} \left| \mathcal{G}(\boldsymbol{\omega}, j) \overline{\mathcal{G}(\boldsymbol{\omega}, j) * \mathcal{P}(\boldsymbol{\omega})} \right| d\boldsymbol{\omega}$$

This concludes the proof.

$\square$

## A.4 Regularisation in Regression Models and Autoencoders

In the case of regression the most commonly used loss is the mean-square error.

$$\mathcal{L}(\mathbf{x}, \mathbf{y}) = \frac{1}{2}(\mathbf{y} - \mathbf{h}_L(\mathbf{x}))^2$$

In this case, $\mathbf{H}_{L,n}$ is $\mathbf{I}$. As such:

$$R(\mathcal{B}; \boldsymbol{\theta}) = \frac{1}{2}\mathbb{E}_{\mathbf{x} \sim \mathcal{B}}\left[\sum_{k=0}^{L} \sigma_k^2 (\text{Tr}(\mathbf{J}_k(\mathbf{x})^\intercal \mathbf{J}_k(\mathbf{x})))\right] = \frac{1}{2}\mathbb{E}_{\mathbf{x} \sim \mathcal{B}}\left[\sum_{k=0}^{L-1} \sigma_k^2 (\|\mathbf{J}_k(\mathbf{x})\|_F^2)\right]$$

This added term corresponds to the trace of the covariance matrix of the outputs $\mathbf{h}_L$ given an input $\mathbf{h}_k$. As such we are penalising the sum of output variances of the approximator; we are penalising the sensitivity of outputs to perturbations in layer $k$ (Webb, 1994; Bishop, 1995).

For ReLU-like activations (ELU, Softplus ...) , because our functions are at *most* linear, we can bound our regularisers using the Jacobian of an equivalent linear network:

$$\sum_{k=0}^{L} \sigma_k^2 (\|\mathbf{J}_k(\mathbf{x})\|^2) < \sum_{k=0}^{L} \sigma_k^2 (\|\mathbf{J}_k^{\text{linear}}(\mathbf{x})\|^2) = \sum_{k=0}^{L} \sigma_k^2 (\|\mathbf{W}_L \dots \mathbf{W}_k\|^2)$$

Where $\mathbf{J}_k^{\text{linear}}(\mathbf{x})$ is the gradient evaluated with no non-linearities in our network. This upper bound is reminiscent of $rank - k$ ridge regression, but here we penalise each sub-network in our network (Kunin et al., 2019). Also note that the regression setting is directly translatable to Auto-Encoders, where the labels are the input data.

## A.5 Regularisation in Classifiers

In the case of classification, we consider the cross-entropy loss. Recall that we consider our network outputs $\mathbf{h}_L$ to be the pre-softmax of logits of the final layer $\mathbf{L}$. We denote $\mathbf{p}(\mathbf{x}) = \text{softmax}(\mathbf{h}_L(\mathbf{x}))$. The loss is thus:

$$\mathcal{L}(\mathbf{x}, \mathbf{y}) = -\sum_{c=0}^{M} \mathbf{y}_{n,c} \log(\text{softmax}(\mathbf{h}_L(\mathbf{x}))_c) \tag{13}$$

where $c$ indexes over the $M$ possible classes of the classification problem. The hessian $\mathbf{H}_L$ in this case is easy to compute and has the form:

$$\mathbf{H}_L(\mathbf{x})_{i,j} = \begin{cases} \mathbf{p}(\mathbf{x})_i(1 - \mathbf{p}(\mathbf{x})_j) & i = j \\ -\mathbf{p}(\mathbf{x})_i \mathbf{p}(\mathbf{x})_j & i \neq j \end{cases} \tag{14}$$

As Wei et al. (2020), Sagun et al. (2018), and LeCun et al. (1998) show, this Hessian is PSD, meaning that $\text{Tr}(\mathbf{J}_k \mathbf{H}_L \mathbf{J}_k^\intercal)$ will be positive, fulfilling the criteria for a valid regulariser.

$$R(\mathcal{B}; \boldsymbol{\theta}) = \mathbb{E}_{\mathbf{x} \sim \mathcal{B}}\left[\frac{1}{2}\sum_{k=0}^{L} \sigma_k^2 \sum_{i,j} (\mathbf{H}_L(\mathbf{x}) \circ \mathbf{J}_k(\mathbf{x})\mathbf{J}_k^\intercal(\mathbf{x}))_{i,j}\right]$$

$$= \mathbb{E}_{\mathbf{x} \sim \mathcal{B}}\left[\frac{1}{2}\sum_{k=0}^{L} \sigma_k^2 \sum_{i,j} (\text{diag}(\mathbf{H}_L(\mathbf{x}))^\intercal \mathbf{J}_k^2(\mathbf{x}))_{i,j} + \frac{1}{2}\sum_{k=0}^{L} \sigma_k^2 \sum_{\forall i,j \ i \neq j} (\mathbf{H}_L(\mathbf{x}) \circ \mathbf{J}_k(\mathbf{x})\mathbf{J}_k^\intercal(\mathbf{x}))_{i,j}\right]$$

$\text{diag}(\mathbf{H}_L(\mathbf{x}))^\intercal$ is the row vector of the diagonal of $\mathbf{H}_L(\mathbf{x})$. The first equality is due to the fact that $\mathbf{H}_L$ is symmetric and is due to the commutative properties of the trace operator. The final equality is simply the decomposition of the sum of the matrix product into diagonal and off-diagonal elements. For shallow networks, the off-diagonal elements of $\mathbf{J}_k \mathbf{J}_k^\intercal$ are likely to be small and it can be approximated by $\mathbf{J}_k^2$ (Poole et al., 2016; Hauser and Ray, 2017; Farquhar et al., 2020; Aleksziev, 2019). See Figure A.1 for a demonstration that the off-diagonal elements of $\mathbf{J}_k^\intercal \mathbf{J}_k$, are negligible for smaller networks. Ignoring these off-diagonal terms, we obtain an added positive term:

$$R(\mathcal{B}; \boldsymbol{\theta}) \approx \mathbb{E}_{\mathbf{x} \sim \mathcal{B}}\left[\frac{1}{2}\sum_{k=0}^{L} \sigma_k^2 \sum_{i,j} (\text{diag}(\mathbf{H}_L(\mathbf{x}))^\intercal \mathbf{J}_k^2(\mathbf{x}))_{i,j}\right] \tag{15}$$

For ReLU-like activations (ELU, Softplus ...), because our functions are at *most* linear, we can bound our regularisers using the Jacobian of an equivalent linear network:

$$\sum_{k=0}^{L} \sigma_k^2 \sum_{i,j} (\text{diag}(\mathbf{H}_L(\mathbf{x}))^\intercal \mathbf{J}_k(\mathbf{x})^2)_{i,j} < \sum_{k=0}^{L} \sigma_k^2 \sum_{i,j} (\text{diag}(\mathbf{H}_L(\mathbf{x}))^\intercal (\mathbf{W}_L \dots \mathbf{W}_k)^2))_{i,j} \quad (16)$$

(a) SVHN MLP, $k$=0     (b) SVHN MLP, $k$=1     (c) SVHN MLP, $k$=2

(d) CIFAR10 CONV, $k$=0    (e) CIFAR10 CONV, $k$=1    (f) CIFAR10 CONV, $k$=2

Figure A.1: Samples of heatmaps of 10 by 10 matrices $\mathbf{J}_k^\intercal \mathbf{J}_k$ ($k$ indexing over layers) for 2-layer MLPs and convolutional networks (CONV) trained to convergence (with no regularisation) on the SVHN and CIFAR10 classification datasets, each with 10 classes. We can clearly see that the diagonal elements of these matrices dominate in all examples, though less so for the data layer.

# B   Tikhonov Regularisation

Note that because we are penalising the terms of the Sobolev norm associated with the first order derivatives, this constitutes a form of Tikhonov regularisation. Tikhonov regularisation involves adding some regulariser to the loss function, which encodes a notion of 'smoothness' of a function $f$ (Bishop, 1995). As such, by design, regularisers of this form have been shown to have beneficial regularisation properties when used in the training objective of neural networks by smoothing the loss landscape (Girosi and Poggio, 1990; Burger and Neubauer, 2003). If we have a loss of the form $\mathcal{L}(\mathcal{B}; \boldsymbol{\theta})$, the Tikhonov regularised loss becomes:

$$\mathcal{L}(\mathcal{B}; \boldsymbol{\theta}) + \lambda \|f^\theta\|_{\mathcal{H}}^2 \tag{17}$$

where $f^\theta$ is the function with parameters $\theta$ which we are learning and $\|\cdot\|_{\mathcal{H}}$ is the norm or semi-norm in the Hilbert space $\mathcal{H}$ and $\lambda$ is a (multidimensional) penalty which penalises elements of $\|f^\theta\|_{\mathcal{H}}^2$ unequally, or is data-dependent (Tikhonov, 1977; Bishop, 1995). In our case $\mathcal{H}$ is the Hilbert-Sobolev space $W_\mu^{1,2}(\mathbb{R}^d)$ with norm dictated by Equation (7). $R(\cdot)$ penalises the function's semi-norm in this space.

# C   Measuring Calibration

A neural network classifier gives a prediction $\hat{y}(\mathbf{x})$ with confidence $\hat{p}(\mathbf{x})$ (the probability attributed to that prediction) for a datapoint $\mathbf{x}$. Perfect calibration consists of being as likely to be correct as you are confident:

$$p(\hat{y} = y | \hat{p} = r) = r, \quad \forall r \in [0, 1] \tag{18}$$

To see how closely a model approaches perfect calibration, we plot reliability diagrams (Guo et al., 2017; Niculescu-Mizil and Caruana, 2005), which show the accuracy of a model as a function of its confidence over $M$ bins $B_m$.

$$\mathrm{acc}(B_m) = \frac{1}{|B_m|} \sum_{i \in B_m} \mathbf{1}(\hat{y}_i = y_i) \tag{19}$$

$$\mathrm{conf}(B_m) = \frac{1}{|B_m|} \sum_{i \in B_m} \hat{p}_i \tag{20}$$

We also calculate the Expected Calibration Error (ECE) Naeini et al. (2015), the mean difference between the confidence and accuracy over bins:

$$\mathrm{ECE} = \sum_{m=1}^{M} \frac{|B_m|}{N} |\mathrm{acc}(B_m) - \mathrm{conf}(B_m)| \tag{21}$$

However, note that ECE only measures calibration, not refinement. For example, if we have a balanced test set one can trivially obtain $\mathrm{ECE} \approx 0$ by sampling predictions from a uniform distribution over classes while having very low accuracy.

# D   Classification Margins

Typically, models with larger classification margins are less sensitive to input perturbations (Sokolić et al., 2017; Jakubovitz and Giryes, 2018; Cohen et al., 2019; Liu et al., 2019; Li et al., 2018). Such margins are the distance in data-space between a point $\mathbf{x}$ and a classifier's decision boundary. Larger margins mean that a classifier associates a larger region centered on a point $\mathbf{x}$ to the same class. Intuitively this means that noise added to $\mathbf{x}$ is still likely to fall within this region, leaving the classifier prediction unchanged. Sokolić et al. (2017) and Jakubovitz and Giryes (2018) define a classification margin $M$ that is the radius of the largest metric ball centered on a point $\mathbf{x}$ to which a classifier assigns $\mathbf{y}$, the true label.

**Proposition 1** (Jakubovitz and Giryes (2018)). *Consider a classifier that outputs a correct prediction for the true class A associated with a point $\mathbf{x}$. Then the first order approximation for the l2-norm of the classification margin $M$, which is the minimal perturbation necessary to fool a classifier, is lower bounded by:*

$$M(\mathbf{x}) \geq \frac{(\mathbf{h}_L^A(\mathbf{x}) - \mathbf{h}_L^B(\mathbf{x}))}{\sqrt{2}\|\mathbf{J}_0(\mathbf{x})\|_F}. \tag{22}$$

|                  |                 |
| :--------------: | :-------------: |
| (a) $\mathbf{J}_0$ CIFAR | (b) $\mathbf{J}_0$ SVHN |

Figure D.2: Here we show distribution plots of $\mathbf{J}_0$ for 2-layer MLPs trained on CIFAR10 (a) and SVHN (b) for models trained with no noise (Baseline), models trained with noise on their inputs (GNI Input), models trained with noise on all their layers (GNI All Layers). Noising all layers induces a larger penalisation on the norm of $\mathbf{J}_0$, seen clearly here by the shrinkage to 0 of $\mathbf{J}_0$ for models trained in this manner.

*We have $\mathbf{h}_L^A(\mathbf{x}) \geq \mathbf{h}_L^B(\mathbf{x})$, where $\mathbf{h}_L^A(\mathbf{x})$ is the $L^{th}$ layer activation (pre-softmax) associated with the true class A, and $\mathbf{h}_L^B(\mathbf{x})$ is the second largest $L^{th}$ layer activation.*

Networks that have lower-frequency spectrums and consequently have smaller norms of Jacobians (as established in Section 4 ), will have larger classification margins and will be less sensitive to perturbations. This explains the empirical observations of Rahaman et al. (2019) which showed that functions biased towards lower frequencies are more robust to input perturbations.

What does this entail for GNIs applied to each layer of a network ? We can view the penalisation of the norms of the Jacobians, induced by GNIs for each layer $k$, as an unweighted penalisation of $\|\mathbf{J}_0(\mathbf{x})\|_F$. By the chain rule $\mathbf{J}_0$ can be expressed in terms of any of the other network Jacobians $\mathbf{J}_0(\mathbf{x}) = \mathbf{J}_k(\mathbf{x})\frac{\partial \mathbf{h}_k}{\mathbf{x}} \forall k \in [0 \ldots L]$. We can write $\|\mathbf{J}_0(\mathbf{x})\|_F = \|\mathbf{J}_k(\mathbf{x})\frac{\partial \mathbf{h}_k}{\mathbf{x}}\|_F \leq \|\mathbf{J}_k(\mathbf{x})\|_F \|\frac{\partial \mathbf{h}_k}{\mathbf{x}}\|_F$. Minimising $\|\mathbf{J}_0(\mathbf{x})\|_F$ is equivalent to minimising $\|\mathbf{J}_k(\mathbf{x})\|_F$ and $\|\frac{\partial \mathbf{h}_k}{\mathbf{x}}\|_F$, and upweighted penalisations of $\|\mathbf{J}_k(\mathbf{x})\|_F$ should translate into a shrinkage of $\|\mathbf{J}_0(\mathbf{x})\|_F$. As such, noising each layer should induce a smaller $\|\mathbf{J}_0(\mathbf{x})\|_F$, and larger classification margins than solely noising data. We support this empirically in Figure D.2.

In Figure F.6 we confirm that these larger classification translate into a lessened sensitvity to noise.

## E    Model Capacity

Intuitively one can view lower frequency functions as being 'less complex', and less likely to overfit. This can be visualised in Figure 4. A measure of model complexity is given by 'capacity' measures. If we have a model class $\mathcal{H}$, then the capacity assigns a non-negative number to each hypothesis in the model class $\mathcal{M} : \{\mathcal{H}, \mathcal{D}_{train}\} \to \mathbb{R}^+$, where $\mathcal{D}_{train}$ is the training set and a lower capacity is an indicator of better model generalisation (Neyshabur et al., 2017). Generally, deeper and narrower networks induce large capacity models that are likely to overfit and generalise poorly (Zhang et al., 2017). The network Jacobian's spectral norm and Frobenius norm are good approximators of model capacity and are clearly linked to $R$ (Guo et al., 2017; Neyshabur et al., 2017, 2015).

The Frobenius norm of the network Jacobian corresponds to a norm in Sobolev space which is a measure of a network's high-frequency components in the Fourier domain. From this we offer the first theoretical results on why norms of the Jacobian are a good measure of model capacity: as low-frequency functions correspond to smoother functions that are less prone to overfitting, a smaller norm of the Jacobian is thus a measure of a smoother 'less complex' model.

# F   Additional Results

(a) BHP MLP Loss

Figure F.3: In Figure (a) we show the test set loss for the regression dataset Boston House Prices (BHP) for 4-layer ELU MLPs trained with $R(\cdot)$ and GNIs for $\sigma^2 = 0.1$. We compare to a non-noised baseline (Baseline). Exp Reg captures much of the effect of noise injections. The test set loss is quasi-identical between Exp Reg and Noise runs which clearly differentiate themselves from Baseline runs.

(a) SVHN MLP, $\sigma^2 = 0.1$          (b) BHP MLP $\sigma^2 = 0.1$

Figure F.4: Here we use small variance noise injections and show that the $R(\cdot)$ (Exp Reg) in equation (11) and (14), induces the same trajectory through the loss landscape as GNIs (Noise). We show the trace of the Hessian of neural weights ($H_{i,j} = \frac{\partial \mathcal{L}}{\partial w_i \partial w_j}$) for a smaller 2-layer 32 unit MLP trained on the classification datasets CIFAR10 (a), and SVHN (b), and the regression dataset Boston House Prices (BHP) (c). In all experiments we compare to a non-noised baseline (Baseline). $\text{Tr}(\mathbf{H})$, which approximates the trajectory of the model weights through the loss landscape, is quasi identical for Exp Reg and Noise and is clearly distinct from Baseline, supporting the fact that the explicit regularisers we have derived are valid. As expected the explicit regulariser and the noised models have smoother trajectories (lower trace) through the loss landscape, except for CIFAR10.

(a) ELU non-linearities, $\sigma^2 = 0.1$

(b) ReLU non-linearities, $\sigma^2 = 0.1$

Figure F.5: Illustration of the loss induced by the $R(\cdot)$ for classification detailed in equation (14) for convolutional and MLP architectures, and for ReLU and ELU non-linearities. The loss trajectory is quasi-identical to models trained with GNIs and the trajectories are clearly distinct from baselines (Baseline), supporting the fact that the explicit regularisers we have derived are valid.

(a) CIFAR  (b) SVHN

Figure F.6: In (a) and (b) a model's sensitivity to noise by adding noise of variance $\alpha^2$ to data and measuring the resulting model accuracy given this corrupted test data. We show this for 2-layer MLPs trained on CIFAR10 (a) and SVHN (b) for models trained with no noise (Baseline), models trained with noise on their inputs (GNI Input), models trained with noise on all their layers (GNI All Layers), and models trained with the $R(\cdot)$ for classification. Noise added during training has variance $\sigma^2 = 0.1$ and confidence intervals are the standard deviation over batches of size 1024. Models trained with noise on all layers, and those trained with $R(\cdot)$, have the slowest decay of performance as $\alpha$ increases, confirming that such models have larger classification margins.

(a) CIFAR10 MLP, $\sigma = 0.1$      (b) CIFAR10 MLP, $\sigma = 0.1$

(c) SVHN MLP, $\sigma = 0.1$      (d) SVHN MLP, $\sigma = 0.1$

(e) CIFAR10 CONV, $\sigma = 0.1$      (f) CIFAR10 CONV, $\sigma = 0.1$

(g) SVHN CONV, $\sigma = 0.1$      (h) SVHN CONV, $\sigma = 0.1$

Figure F.7: Illustration of how Gaussian noise (Noise) *additions* improve calibration relative to models trained without noise injections (Baselines) and how $R(\cdot)$ (Exp Reg) also captures some of this improvement in calibration. We include results for MLPs and convolutional networks (CONV) with ELU activations on SVHN and CIFAR10 image datasets. On the left hand side we plot reliability diagrams (Guo et al., 2017; Niculescu-Mizil and Caruana, 2005), which show the accuracy of a model as a function of its confidence over $M$ bins $B_m$. Models that are perfectly calibrated have their accuracy in a bin match their predicted confidence: this is the dotted line appearing in figures. We also calculate the Expected Calibration Error (ECE) which measures a model's distance to this ideal (see Appendix C for a full description of ECE) (Naeini et al., 2015). Clearly, Noise and Exp Reg models are better calibrated with a lower ECE relative to baselines. This can also be appraised visually in the reliability diagram. The right hand side supports these results. We show density plots of the entropy of model predictions. One-hot, highly confident, predictions induce a peak around 0, which is very prominent in baselines. Both Noise and Exp Reg models smear out predictions, as seen by the greater entropy, meaning that they are more likely to output lower-probability predictions.

# G Network Hyperparameters

All networks were trained using stochastic gradient descent with a learning rate of 0.001 and a batch size of 512.

All MLP networks, unless specified otherwise, are 2 hidden layer networks with 512 units per layer.

All convolutional (CONV) networks are 2 hidden layer networks. The first layer has 32 filters, a kernel size of 4, and a stride length of 2. The second layer has 128 filters, a kernel size of 4, and a stride length of 2. The final output layer is a dense layer.