[Reviews · NeurIPS 2020]

Review 1

Summary and Contributions: The paper proves a theorem about the expected change in loss of a neural network caused by introducing Gaussian noise at every layer. The paper uses this theorem to show that adding this expected additional loss can be viewed as Tikhonov regularization on the Sobolev norm of the function the network computes.

Strengths: If the additional loss due to Gaussian noise injection really is well described by equation (10) in practice, it would provide a nice mathematical lens for studying the regularizing properties of noise injection.

Weaknesses: ... Unfortunately, I am not at all convinced that equation (10) reflects what happens in practice. Equation 10 is obtained by linearizing the neural network around its input to get a first-order approximation of the effect of the noise on the output layer, then determining the additional loss that this first-order effect produces. The equation includes a $O(\kappa)$ term which "disappears in the limit of small noise". The trouble is, in a deep network, each layer may compound the higher-order effects of changes in lower layers. So, it is not clear to me at all that gaussian noise injection in practice is small enough for these higher order effects to be ignored. EDIT: To elaborate, I feel that the experimental results in the paper are not adequately explained by the theory they present. To make a specific complaint, the authors state on line 116 that "for the rest of this work, we assume that noise injections are of small variance and that as such, the higher order terms of Proposition B and Theorem 1 can be ignored." Then, in the experiments described in Figure 1 and Figure 2, they compare explicitly regularized networks with networks trained with noise variance \sigma^2 = 0.1. I don't think there is any justification in the paper for why this value of sigma^2 should be small enough that the higher order terms can be ignored as the authors seem to imply. The plots indicate that the Gaussian noise injection has some of the same effects as explicit regularization, but maybe these are just properties that come with any natural scheme for regularizing a neural network, and don't actually have to do with the regularizer derived in Theorem 1.

Correctness: The theorem seems to be correct.

Clarity: The paper is clearly written.

Relation to Prior Work: The paper discusses prior work on regularization and explains how the main theorem can be applied to these other results.

Reproducibility: Yes

Additional Feedback: My feeling is that for this paper to be an accept, it would have to analyze in much greater depth how the size of the higher order terms depend on $\epsilon_k$ and on the activation function EDIT: To provide something specific, perhaps something you could do to justify the efficacy of theorem 1 would be to compare the size of the O(kappa) term with the sigma^2 Tr(JHJ) term for the experiments they do, and show the latter is always less than the former (I don't know if this will be the case, though). A small error: In proposition 1, you have the term $\epsilon_k J_k(x)$, where $\epsilon_k$ is the noise at layer $k$, and should be a vector of length $N_k$, and $J_k(x)$ is the Jacobian, a $N_L \times N_k$ matrix. This should be $J_k(x)\epsilon_k$ as it is in equation (5) of the appendix. EDIT 2 8/14/2020 The authors response included a plot of the error term that appears in their equations, and showed that it is relatively small with respect to the main term. I'll admit my intuition somehow told me this would not be the case, but now that I see it is, I think the theorem is a nice lens for the relation between these two forms of regularization. In light of this, and pending more discussion with the authors, I'm updating my score to accept.


Review 2

Summary and Contributions: This work analyzes the explicit regularization effects of Gaussian Noise Injection (GNI) and connects it to classic function space regularization methods like Tikhonov regularization. GNI considers a neural network and injects Gaussian noise into every layer before evaluating the function. This `smooths out' the output of the function and intuitively regularizes it. When the added noise is `small enough', the authors show that in expectation this gives a new `smoothed' loss function which consists of two parts: 1. the original loss, 2. Additional term depending on the noise variance -- which is the `regularizer'. The first order term of the Taylor series expansion of the `regularizer' function is derived under assumption of small variance. This roughly corresponds to the squared Frobenius norm of various `Jacobians' corresponding to the network. By the fundamental relationship between polynomial multiplication in Fourier domain and derivatives of the given function, these regularizers can be seen to be penalizing the high frequency components in the output function. In the empirical study regarding learning a mixture of low frequency and high frequency sinusoids, the GNI and explicitly regularized networks seems to learn the dominant low frequencies whereas the baseline network overfits the data. This shows that 1. GNI and the explicit regularization based on first order taylor expansion behave similarly in practice. 2. GNI and regularization cut away high frequency components. The work also includes experiments which show that upon adding noise to the data, the performance of networks trained with GNI or explicit regularization decays slower than that of the baseline network.

Strengths: 1. This gives a very intuitive and sensible explanation for explicit regularization via noise injection. 2. The arguments are very simple to understand and agrees with the empirical evaluation in this work and previous results. 3. The contribution seems to be novel and relevant to the NeurIPS community. ======== The authors have addressed most of my concerns with the rebuttal. I maintain my score.

Weaknesses: 1. The work is neither fully rigorous nor empirical. There are a lot of \approx used and higher order terms are left out. 2. The authors try to fill this gap using empirical evaluations but since such empirical results are already established, it does not seem to be novel. There is a lot of gaps in reasoning when jumping between topics like explicit regularization to margins.

Correctness: The claims and methods seem to be correct to the best of my knowledge. I am unaware of Plancherel-Parseval formula given in Equation 16 - when R^d is replaced by a subset Omega. I checked the cited work - Novak et. al and it dealt only with the case Omega = R^d. The authors need to give a correct reference for this formula.

Clarity: It is my opinion that the paper is a bit badly written. A lot of quantities like p_w(y|x) are left undefined. There seem to be a lot of abrupt jumps in the paper and the main body does not fully explain how Section n is connected to the previous sections - this is especially true with respect to Section 5 where the work becomes almost purely empirical. I would suggest that the authors remove some of the later sections so that they can expand and explain the theoretical sections more clearly. Minor quibs: 1. The description of penalization below (7) doesn't seem to make sense. What does penalize the elements of \|f\|^2 mean? 2. What is `x' in proposition 1? Are these elements of the mini-batch? 3. Equation (17) should be "\approx" and (18) should be exact equality.

Relation to Prior Work: Yes, to the best of my knowledge.

Reproducibility: Yes

Additional Feedback:


Review 3

Summary and Contributions: The paper analyzes the effect of Gaussian noise injection at the level of activations within a feed-forward neural network. The authors formally describe the effect of such noise injection, tie it to regularization in the Fourier domain and Tikhonov regularization, and empirically show that noise injection improves classification margins and calibration.

Strengths: The topic considered is very interesting and additional work in this area is necessary. The theoretical analysis in the paper appears sound, and is tied to empirical demonstrations of the relevance of noise injection for classification margins and calibration.

Weaknesses: The theoretical results do not seem to be especially difficult (but they are useful). The conclusions drawn are also not especially surprising (but a grounding for them is nonetheless important). I would love to see more experiments comparing the addition of noise at activations to other forms of regularization, including the addition of noise to the data instead (this last is done in Fig. 3a but not in other experiments).

Correctness: Overall the conclusions seem to be correct. Are Figures 3 and 4 evaluated over training or testing examples? In Figure 4(a), is the confidence simply averaged over many samples? In 4(b), how can there be negative values of the entropy? The x-axis should presumably be cut off at 0, and there should be no artificial smoothing of the curve.

Clarity: In equation (3), the notation of \circ could perhaps be changed to another symbol, since \circ generally denotes composition of functions. In equation (6), \tilde{L} is never actually defined, so this is somewhat confusing. Also, the negative sign on the RHS has been omitted. The notation \mathcal{H} is reused for function space and entropy. I would recommend changing one of these to avoid confusion. It might be clearer to put the Related Work section earlier in the paper, for example after the intro. In line 122, the regularization R does not seem to be defined. Typo ("unpweighted") in line 216. In Figure 3, the caption and x-axis should make clear it is the Frobenius norm of J_0. In Figure 4, is there a reason for not combining the plots in (a) and (b)? In particular, (b) is misleading since the x-axis is scaled differently between the two plots.

Relation to Prior Work: Comparison to related work seems to be clear.

Reproducibility: Yes

Additional Feedback:


Review 4

Summary and Contributions: The paper provides a theoretical analysis of the regularization effects of adding Gaussian noise in the intermediate layers of neural networks during training. The authors show that for sufficiently small values of the noise std, this is mathematically equivalent to a form of Tikhonov regularization. A small set of experiments are provided confirming that the theory gives accurate predictions for small networks on SVHN and CIFAR-10.

Strengths: The paper is well written, and the provided theory provides some welcome insights into what was previously a primarily adhoc regularization technique. The authors provide code for replicating their experiments (I have no tried to run it, but the README and code seems quite detailed). The theory is to the best of my knowledge new.

Weaknesses: I felt the authors could have tested the limits of the theory more. For example, it would be nice to include an empirical analysis to determine what noise range the main theorems hold, currently it is only discussed that they hold for "sufficiently small sigma". Plotting the error term for Proposition 1 as a function of sigma for several models would give a better picture here. Even if the theory doesn't give tight bounds for larger sigma, it's still quite interesting so I encourage the authors to not be shy in testing the limits here. The connection to Fourier Domain Regularization section is interesting, though I wasn't sure why it's significant. What implications are there for this equivalence that's not already implied by the results in Section 3? This could be discussed further. More experimental details could be provided in the Appendix. Currently, not enough information is provided to reproduce the main plots in the paper (e.g. exact architecture specs, learning rate, optimization, batch size ect.). Adding exact run commands for reproducing experiments in the code README would be even better. Most of the experiments were run on small models, and not more modern architectures.

Correctness: Though I did not have time to read the proofs closely, skimming the results I was not able to identify any flaws.

Clarity: This is one of the strengths of the paper.

Relation to Prior Work: Not super familiar with prior theoretical work to comment in much detail on this. Certainly the analysis adds something meaningful to the 2014 Poole et. al. paper.

Reproducibility: Yes

Additional Feedback: 2.1.1, I'd add some citations to discuss the brittleness of neural networks to even slight distribution shift. e.g. https://arxiv.org/abs/1807.01697. The noise robustness (particularly high frequency noise) benefits of larger margin classifiers are clear, however worth mentioning that the sometimes comes with a tradeoff on low frequency noise/corruptions. See for example: https://arxiv.org/abs/1906.08988. Section 3: What does it mean for a scalar valued regularization term in the loss to be PSD? Figure 1: What learning rate was the baseline trained on? This blowup in the Hessian trace is typically observed with small learning rates. You may find rerunning this same experiment with large learning rate will make the baseline more similar to the regularization term. Have you tried this? It's still not well understood if regularization methods such as noise injections have orthogonal benefits to other methods such as large learning rate or small batch size. In its current form I'm giving the paper a 6, though I anticipate it would be much stronger is some of my concerns are addressed. In the impact statement you write that regularization can help make the model more robust to adversarial attack. I strongly disagree with this statement, all models are absolutely broken in the presence of any realistic adversary. Improving lp-robustness will do little to mitigate these issues. I'd instead focus on the potential gains on ood robustness of larger margin classifiers (e.g. better robustness to noise).

[Author Response · NeurIPS 2020]

We thank the reviewers for their insights, which we have incorporated into
our work. We were pleased that reviewers highlighted the novelty of our theory
(**R1**,**R2**,**R3**,**R4**), its clarity (**R1**,**R4**), its correctness (**R2**,**R3**,**R4**), and its relevance
and potential impact for the research community (**R1**,**R2**,**R3**,**R4**).

To bolster our claims, we have run additional experiments that support our
approximations, which ignored $O(\kappa)$ in Theorem 1 for 0.1 variance GNIs. In Fig
1 we show the estimation error of the true noisy loss this approximation induces
for 12-layer MLP sigmoid networks for which the interaction effects described
by **R1** would be most prominent. Clearly the error is small for the variance we
used, which is highlighted in red. This experiment strengthens our arguments
and we thank **R1** and **R4** for suggesting it.

To further support this result, we ran experiments which demonstrate that the *Exp*
*Reg* terms (first term of Theorem 1 used in our approximations) dominates $O(\kappa)$
in value for a range of small and large $\sigma^2$ (see Fig 2). The findings of Figs 1, 2
also hold for ReLU and ELU activations and we will include these results in the
update. Combined, these new results show that our approximations hold even for
large variance GNIs, making our work even more impactful.

**Reviewer 1**: We thank the reviewer for their insights and as seen in the results
above, they have already stimulated new experiments which strengthen our
arguments. To further alleviate your concerns, we have run experiments which
demonstrate that the regularisation induced by *Exp Reg* (first term of Theorem
1) matches that of GNIs and is not just a byproduct of any generic regularisation
method. See Fig 3 for a demonstration of this. We also thank you for pointing
out the typo in Proposition 1.

**Reviewer 2**: The restriction of Plancherel's Theorem to a compact subset is
straightforward, and we will improve the clarity surrounding this, as suggested.
We will also make sure to include the justification for this in the Appendix. We
will also clarify the connection between Sec.4, 5. **Note**: 'Elements of $\|f\|^2$'
below Eq (7) should read as the 'constituent terms of $\|f\|^2$'. Take the Sobolev
norm for instance, we could penalise the $L_p$ norm of the function and the $L_p$
norm of its derivative with different weightings. $\mathbf{x}$ in Proposition 1 is elements
of the minibatch; we'll clarify this.

**Reviewer 3**: Our theory encompasses the case of injecting noise solely on data,
as this corresponds to layer 0 in our formulation. This translates into a lessened
penalisation of the network Jacobians. In particular, we ran experiments following
this review which showed that the dampening of higher-order frequencies in the
Fourier domain is less when injecting noise only on data. We will include these
results in our update. See also **R1** and Fig 3 for a comparison of GNIs to L2
regularisation. **Note**: We thank you for your detailed review of Figs 3 and 4. We
will improve the figures given your feedback.

**Reviewer 4**: We thank the reviewer for providing such detailed comments. As
seen above, we have run experiments testing the limits of our assumptions (see
also **R1**). As our theory is in early stages we wanted to test its efficacy primarily
on simpler models. We are working on scaling it to larger models, eg VGG13:
currently the calculation of each layer's Jacobian is memory hungry, as is $\text{Tr}(\mathbf{H})$
in Fig 1 [paper]. We thank you for pointing us to works on the brittleness of neural
networks and on the tradeoffs between robustness to noise vs. other types of data
corruption. We will include these in our update along with with more detailed
commands in the README to replicate experiments. We will also clarify the
connection between Secs 4 & 5. **Note**: A "PSD" scalar is incorrect as you have
pointed out, it should read that the "constituent terms" of the scalar are PSD.

You make an interesting point about Fig 1: All models were trained with a relatively low learning rate (lr) of 0.001,
supporting your claim of low lr Hessian 'blowup'. In light of this we have run the baseline with lr=0.1 and found that
$\text{Tr}(\mathbf{H})$ *decreases* with training instead. As you suggest, there could be a lr for which we recover some of the benefits of
GNIs. Exploring this connection further would be a very interesting stream of research.

Figure 1: Proportional estimation error (maximum of 1.0) of $\mathbb{E}[\tilde{\mathcal{L}}_{\text{SGD}}(\mathcal{B};\boldsymbol{\theta},\boldsymbol{\epsilon})]$ when ignoring $O(\kappa)$ from Theorem 1. We plot this for 12-layer MLPs trained on CIFAR10 with sigmoid activations, non-linearities and network depth which heavily test our approximations. GNIs are applied to each layer bar the final layer. Even under these conditions, 0.1 variance GNIs (highlighted in red) are well approximated by our estimations. Shading is the standard deviation over 250 points.

Figure 2: Here we plot *Exp Reg*, the first term of Theorem 1, against $O(\kappa)$ for $\sigma^2 \in [0.1, 0.25, 1.0, 4.0]$. Networks are the same as in Fig 1. For reference we plot $y = x$ in red. For all values of $\sigma^2$, *Exp Reg* lies above this line and we can claim that empirically *Exp Reg* $> O(\kappa)$.

Figure 3: Training set loss for 4-layer ELU MLPs trained on SVHN. Three regularisation methods are compared: $L_2$ regularisation with $\lambda =0.01$ penalisation, 0.1 variance GNIs, and *Exp Reg*, the first term of Theorem 1. $\lambda$ was picked such that the magnitude of the regularisation roughly matched that of GNIs/Exp Reg at the start of training. Clearly GNIs/Exp Reg have distinct training curves to L2.

[Meta-Review · NeurIPS 2020]

The paper provides a characterization of regularization resulting from Gaussian Noise Injection (GNI) in the limit of infinitesimal noise variance. The contributions are insightful and the results are useful for the community. However, based on reviewers feedback and my own reading, there is lot of scope for improvement 1. The writing/clarity is severely lacking in the submission file - many notations are bad/undefined/overloaded; technical terms used non-rigorously. Along with the points raised by the reviewers, here are some additional comments - theta and w represent the same quantity and used confusingly throughout the paper - looks like two different authors were editing the manuscript without discussing notation - tile{L} & p_w(y|\tilde{h}) undefined) ; - in prop 1 and thm2: O(\gamma) and O(\kappa) can be more intuitively replaced by o(\epsilon) and o(\mathcal{E}_L) - the regularization term as written in eq. 7 does not make sense for multi-dim lambda under standard notation where ||f||_H is a scalar - h is never defined (only \tilde{h} and \hat{h}) yet it’s used liberally in definition of Jacobian in line 102, prop 1, thm 1. Also x undefined in prop 1 - in line 119, what does it mean for the result to be "locally exact"? what happens at the kinks in ReLU (hat{h}_k(x) =0) - Fig 3 caption: for (a) it says sigma^2 =0.1, but the plot itself is about varying alpha 2. While the spectral characterization in eq. 18 is useful, it would be more useful to add a discussion of what the characterization means - specially pointing out the consequences of the fact that all the components of the sums are not independent, e.g. F^k_i is very much related to F^(k-1)_i & F^(k+1)_i - what does it mean in terms of which layer functions are more strongly regularized to have low frequency component. Overall the expressions in eq. 18 and thm 1 are very general equations and it would be useful to make connections to specific architectures/networks, width, depth, etc. even if they are simple networks.